# Effect of gluten-free diet and antibiotics on murine gut microbiota and immune response to tetanus vaccination

**Pernille Kihl[1]☯, Lukasz Krych[2]☯, Ling Deng[2]☯, Lars H. Hansen[3,4,5], Karsten Buschard[6], Søren Skov[1], Dennis S. Nielsen[2], Axel Kornerup Hansen[1]***

**1** Section of Experimental Animal Models, Department of Veterinary and Animal Sciences, University of Copenhagen, Frederiksberg, Denmark, **2** Department of Food Science, University of Copenhagen, Frederiksberg, Denmark, **3** Department of Biology, University of Copenhagen, Copenhagen, Denmark, **4** Department of Environmental Science, Aarhus University, Roskilde, Denmark, **5** Department of Plant and Environmental Sciences, University of Copenhagen, Frederiksberg, Denmark, **6** Bartholin Institute, Rigshospitalet, Copenhagen, Denmark

☯ These authors contributed equally to this work.
* akh@sund.ku.dk

## Abstract

The purpose of this study was to compare the effect of a gluten-free diet and/or antibiotics on tetanus vaccine induced immunoglobulin G titers and immune cell levels in BALB/c mice. The gluten-free diet was associated with a reduced anti-tetanus IgG response, and it increased the relative abundance of the anti-inflammatory *Bifidobacterium* significantly in some of the mice. Antibiotics also led to gut microbiota changes and lower initial vaccine titer. After a second vaccination, neither gluten-free diet nor antibiotics reduced the titers. In the spleen, the gluten-free diet significantly increased regulatory T cell ($T_{reg}$) fractions, $CD4^+$ T cell activation, and tolerogenic dendritic cell fractions and activation, which extend the downregulating effect of the $T_{reg}$. Therefore, the systemic effect of the gluten-free diet seems mainly tolerogenic. Antibiotics reduced the fractions of $CD4^+$ T and B cells in the mesenteric lymph nodes. These results suggest that vaccine response in mice is under influence of their diet, the gut microbiota and the interplay between them. However, a gluten-free diet seems to work through mechanisms different from those induced by antibiotics. Therefore, diet should be considered when testing vaccines in mice and developing vaccines for humans.

## Introduction

Although saving millions of lives worldwide, many vaccines are not universally efficient, and varying or even lacking efficacy poses a major problem in some populations. Multiple reports on the efficacy of vaccines reveal a reduced response in humans from low-middle income countries compared to high income countries [1], which may be due to genetic background, immune status, vitamin status, age and other factors [2–4].

The gut microbiota (GM) is a complex consortium of trillions of microorganisms strongly influencing host health and disease, which has colonized the gut of all mammals [5–9]. A core

**Data Availability Statement:** Raw sequencing data was deposited in the NCBI SRA database (https://www.ncbi.nlm.nih.gov/sra) under the Bioproject PRJNA703942. Clinical and immunological data

are stored at the Open Science Framework https://osf.io/s47np/.

**Funding:** The study was supported by a grant (2013-4) from LIFEPHARM (www.lifepharm.ku.dk) to AKH. The funder had no role in the project.

**Competing interests:** I have read the journal's policy and the authors of this manuscript have the following competing interests: P.K. is today hired by MSD Denmark. A list of potential conflicts of interests for A.K.H. can be seen at https://ivh.ku.dk/english/employees/?pure=en/persons/107126.

GM is established in early life through contact with the mother, early life diet, and extrinsic factors [10]. Later in life, the GM composition and development of both humans and mice is to a large extent influenced by extrinsic factors with diet being among the most important ones [11–13]. Diet with reduced amounts or no gluten changes the GM composition in both humans [14] and mice [15–17]. Gut associated lymphoid tissue (GALT) is an important mediator of GM immune impacts, which is regulated via pattern recognition receptors (PRRs), such as Toll-like receptors (TLRs), and the intracellular nucleotide-binding oligomerization domain, when stimulated by gut microbial associated molecular pattern (MAMPs) [18]. GM communicates with the lungs through the gut-lung axis, which is considered of importance for the risk of developing severe disease in humans infected with respiratory viruses [19]. It has recently been proposed that the low incidence of clinical COVID-19 in some regions may be due to their intake of a mainly gluten-free diet [20].

Therefore, there are strong indications that GM composition influences host immune responses, and, thereby the vaccine activated immune response [21, 22]. Vaccine responses are initiated when naive T cells recognize the antigen associated with Major Histocompatibility Complex (MHC) class II molecules on the surface of antigen presenting cells. Dependent on the microbiota composition gut dendritic cells will secrete IL-12 to favour the development of T helper cells type 1 (Th1) [23], while the TLRs of some specialized tolerogenic dendritic cells induces the formation of regulatory T cells ($T_{reg}$), which regulate the response towards microbes by secreting IL-10 [24]. A gluten-free diet has previously been shown to increase the number of $T_{reg}$ in the gut of mice [25, 26], which have consequences, such as reduced incidence of type 1 diabetes in non-obese diabetic (NOD) mice [15], and reduced risk of islet autoimmunity in humans [27], which are both Th1 dependent. Type 1 diabetes incidence in NOD mice is also reduced by the induction of tolerance to gliadin, which is accompanied by increased proportions of IL-10 positive $T_{reg}$ [26]. As the GM differs essentially between global regions, vaccine response discrepancies between children from low-middle income and high-income countries may be caused by GM composition differences [28]. Children from rural Burkina Faso develop a GM distinctly different from that of Italian children, with the former being enriched in functions well suited to extract energy from the carbohydrate rich diet of rural Burkina Faso [29]. In Bangladeshi infants there is a positive correlation between vaccine responses and the relative abundance of fecal Actinobacteria, which includes *Bifidobacterium* spp. often associated with an anti-inflammatory response [30]. Furthermore, a lower vaccine response was associated with high relative abundances of fecal Clostridiales, Enterobacteriales, and Pseudomonadales [30].

Antibiotics, such as ampicillin, may have a strong impact on GM, which causes decreased relative abundances of $T_{reg}$ and Th in both mice [31] and humans [32]. An early life germ-free period in mice induces long lasting increased activity of CD4$^+$ T cells, i.e. Th1, Th2, and $T_{reg}$ [11], and it is known in mice that antibiotic-driven dysregulation of the GM can modulate immune responses to vaccines in early [33] and in late life [34]. Ampicillin has been hypothesized to block humoral vaccine responses in humans, because its blocking of a specific protein the lack of may induce a transient immunodeficiency [35]. Also, bacterial flagellin stimulation of Toll-like receptor 5 (TLR5) is needed to drive antibody production after influenza vaccination [36]. Neomycin treatment of mice experimentally infected with Influenza A virus increases the disruption of the lung tissue structure and increases lung weight compared to virus infected control mice [37], but local or distal injection of Toll-like receptor (TLR) ligands rescue this neomycin induced immune impairment [38].

We hypothesized that gluten-free diet induced GM perturbations would influence vaccine response in mice. We found it most relevant to study this in relation to a vaccine based upon a bacterial product, and, therefore, we chose to study the impact on a tetanus toxoid vaccine,

although this vaccine generally has high immunogenicity and protection globally [39]. Tetanus toxoid is a bacterial product, which when used as vaccine, is known to raise a strong Th reaction [40], and it has traditionally been used with aluminium oxide hydrate adjuvant, which is expected to drive a Th2 rather than a Th1 response [41]. An increase in $T_{reg}$ relative abundance and subsequent IL-10 secretion, as induced by a gluten-free diet, will reduce both Th1 and Th2 and, thereby antibody producing B cell relative abundance [24, 42]. Therefore, we used the production of specific IgG as primary readout in mice fed a gluten-free or a standard gluten containing diet. To block the impact of the GM, some of the mice were treated or not treated with ampicillin.

## Methods

### Mouse experiments

Experiments were approved by the Animal Experiments Inspectorate, Ministry of Food, Denmark and carried out according to the EU Directive 2010/63/EU on the Protection of Vertebrate Animals used for Experimental and Other Scientific Purposes, and the Danish Animal Experimentation Act (LBK 474 from 15/05/2014). Female BALB/cBomTac mice (Taconic Europe A/S, Ejby, Denmark) health monitored at the breeders and in the experimental facilities, revealing no infections listed in the FELASA guidelines [43], were used. At arrival, they were individually ear marked, weighed, and housed in an AAALAC accredited barrier protected facility. A standard tetanus vaccine for human use (Tetanus vaccine "SSI", D.SP.NR 8767) with 6 LF/ml tetanus toxoid dissolved in aluminium oxide, hydrate ad 1 mg Al/ml, (Statens Seruminstitut, Copenhagen, Denmark) was used. An equal dose of saline (placebo) was administered to the unvaccinated groups. Doses were 0.03 ml (Study A and B) and 0.04 ml (Study C) per mouse given in the neck region. The diets used were either a standard wheat based gluten containing Altromin 1324 diet or a modified Altromin 1334 diet ('Altromin modified') ('Altromin'; Altromin, Lage, Germany), in which wheat protein was replaced with casein as previously described [44]. The mice had free access to drinking water and were all fed *ad libitum* diet. Food consumption was monitored by weight of food administered versus eaten. Ampicillin (Ampivet, Boehringer Ingelhem, Germany) was dosed in the drinking water as 1 gram pr. liter from arrival to euthanasia, while non-treated groups received standard tap water. Upon termination, the total blood volume was collected from the retro-orbital veins into sterile tubes (Eppendorf, Germany), under anaesthesia with fentanyl, fluanisone and midazolam (i.e. 1:1 Hypnorm/Dormicum mixture: 0.315 mg/ml fentanyl + 10 mg/ml fluanisone (VetaPharma, Leeds, UK) and 5 mg/ml midazolam (Roche, Brøndby, Denmark)). At each blood collection, the blood was stored on ice for coagulation for approximately one hour before being centrifuged (500 x g, 10 min). Serum was separated (Micronic tubes) and stored at -20˚C until used for determining of anti-tetanus titers by ELISA. The overall structures of all studies are shown in Fig 1.

**Study A: Impact of gluten on primary and boosting vaccination.** Sixty-three mice, aged six weeks upon arrival, were randomized into three groups of each 21 animals. Two groups were fed Altromin and one group was fed Altromin modified. The Altromin and the Altromin modified group were vaccinated at eight weeks of age and again two and a half weeks after arrival, while the other Altromin group were injected at same time points with placebo (Fig 1A). Body weight of the mice, as well as food consumption was monitored throughout the experiment, and all mice were euthanized and sampled three and a half week after the boosting.

**Study B: Impact of antibiotics on primary and boosting vaccination.** Forty mice, aged five weeks upon arrival, were randomized into two groups of each 20 mice, which were fed Altromin. One group received ampicillin in the drinking water. All mice were vaccinated at

*Study A*

| Group | n | Diet | Antibiotics | Tetanus Vaccine |
|---|---|---|---|---|
| Vaccinated gluten-free | 21 | Altromin modified | No | Yes |
| Vaccinated control | 21 | Altromin | No | Yes |
| Non-vaccinated control | 21 | Altromin | No | Saline |

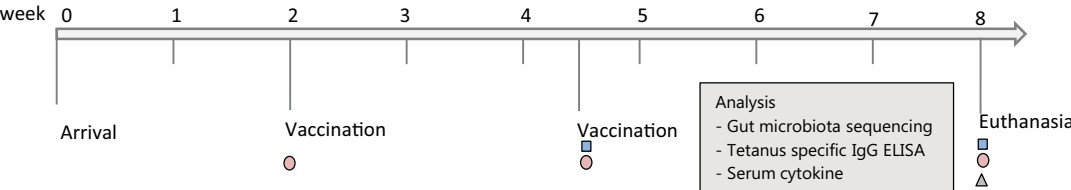

*Study B*

| Group | n | Diet | Antibiotics | Tetanus vaccine |
|---|---|---|---|---|
| Vaccinated antibiotic treated | 20 | Altromin | Yes | Yes |
| Vaccinated control | 20 | Altromin | No | Yes |

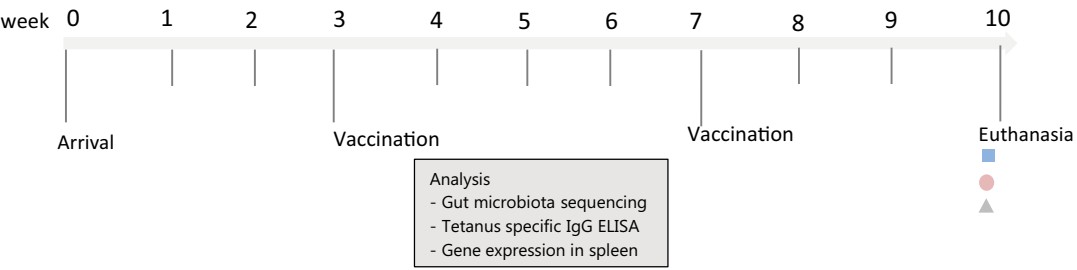

*Study C*

| Group | n | Diet | Antibiotics | Tetanus vaccine |
|---|---|---|---|---|
| Vaccinated gluten-free | 15 | Altromin modified | No | Yes |
| Vaccinated antibiotic treated | 15 | Altromin | Yes | Yes |
| Vaccinated gluten-free antibiotic treated | 15 | Altromin modified | Yes | Yes |
| Vaccinated control | 15 | Altromin | No | Yes |

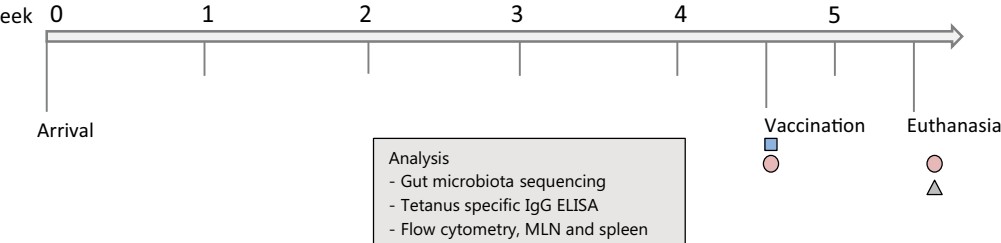

◻ Fecal sample
◯ Blood sample
△ Organ harvest

**Fig 1. Overall study plan for testing vaccine impact from diet and antibiotics.** We set up three studies in BALB/cBomTac mice, to investigate the influence of gut microbiota manipulations on the response to a tetanus vaccine. Two studies (A and B) to study the impact of either a gluten-free diet or antibiotics on the antibody production after a boosted vaccination regime, and one study (C) to study the immediate impact of one vaccination on immune cells counts in gluten-free fed or antibiotics treated mice. In study A we boosted with a second vaccination after two and a half weeks, while we in study B considered that his might be too short and did the boosting after four weeks. In study C we used only one vaccination, because the aim was to see the cell counts in relation to our observations after the first vaccination in study A and B, and, therefore, we also sampled the animals only one week later. Two diets

were used, i.e. the standard wheat based Altromin diet (Altromin), and a modified Altromin diet in which wheat protein was replaced with casein (Gluten-free). 'Antibiotics' indicates that mice were dosed with 1 gram ampicillin per liter drinking water. Control indicates that it is a group on a standard Altromin diet (with gluten) and either vaccinated or not vaccinated. Timelines illustrate the course of the study, and analyses performed.

eight weeks of age and again four weeks later (Fig 1B). Body weight of the mice was monitored upon arrival subsequently throughout the experiment. Body weight of the mice, as well as food consumption was monitored throughout the experiment, and all mice were euthanized and sampled three weeks after the boosting.

**Study C: Antibiotic impact on initial immune functions after primary vaccination.**
Sixty mice aged five weeks at arrival were randomized into four groups of 15 mice per group. Two groups were fed Altromin and two groups were fed Altromin modified, and one Altromin and one Altromin modified group received ampicillin in the drinking water. All groups were vaccinated at nine weeks of age and euthanized and sampled one week later (Fig 1C).

## Gut microbiota characterization

Fecal samples were collected weekly in sterile Eppendorf tubes from arrival until euthanasia, and stored at -80˚C until DNA extraction. Immediately after euthanasia by cervical dislocation, cecum content samples were removed, kept on wet ice and stored at -80˚C. DNA was isolated and extracted from the fecal and cecal samples collected using the DNeasy PowerSoil Kit (# 12888, Qiagen, Germany) following the manufacturer's instructions, but with addition of a bead beating step to increase lysis of bacterial cells. DNA purities and concentrations were determined using a Nanodrop 1000 spectrophotometer (Thermo scientific, USA) prior to being stored at -80˚C. The prokaryotic component of the mouse gut microbiota was characterised by 16S rRNA gene amplicon sequencing. 454 FLX-based 16S rRNA gene sequencing (Study A) was carried out at the National High Throughput DNA Sequencing Centre, University of Copenhagen, Denmark using tag-encoded 454/FLX Titanium (Roche) pyro-sequencing of the V3 and V4 region of the 16S rRNA gene as previously described [45]. NextSeq (Illumina) based 16S rRNA gene (V3-region) amplicon sequencing (Study B and C) was carried out as follows: Total DNA was adjusted to 10 ng/ul, and 5 ul were added to the PCR reaction setups containing 12 ul of Accuprime SuperMix II (ThermoFisher scientific, USA), 1 ul Primer Mix (10 uM nxt_338F `TCGTCGGCAG CGTCAGATGT GTATAAGAGA CAGACWCCTA CGGGWGGCAGCAG` and 10 uM nxt_518R `GTCTCGTGGG CTCGGAGATG TGTATAAGAG ACAGATTACC GCGGCTGCTGG`) and 7 ul of $H_2O$. The V3 region of the 16S rRNA gene was amplified by the following program; 95˚C 2 min, 33 cycles of 95˚C 15 sec, 55˚C x 15 sec and 68˚C x 30 sec, cooled down to 4˚C before the final extension of 4 mins at 68˚C. The amplicons were checked by agarose gel electrophoresis and 2 ul was used for the indexing PCR, if the amplification was successful, PhusionHF Mix (ThermoFisher scientific, USA) and different S5 and S7 primers (20 uM) were combined with the 1st PCR product and went through the following tagging PCR process, 95˚C 2 min, 13 cycles of 95˚C 15 sec, 55˚C 20 sec and 68˚C 20 sec, followed by 5 min final extension at 68˚C. The indexed amplicons were then purified by AMpure XP beads manually or using a Biomek 4000 workstation as described by the manufactures. The final concentrations of the purified amplicons were measured by Qubit HS dsDNA reagents, each product were mixed in equal concentrations (5 ng per sample) and subjected to the Illumina Nextseq550 platform.

## Sequencing data analysis

The raw dataset generated using tag-encoded 454/FLX Titanium (Roche) pyrosequencing were analysed as previously described [46]. The raw data generated by Illumina NextSeq based

amplicon sequencing was processed as previously described [47]. The relative abundance tables generated by both methods were further analysed using QIIME pipeline (V 1.9.1) [48]. Details are provided below. All samples have been normalised by subsampling prior to the analysis setting the subsampling value to the 85% of reads within the most indigent sample (5,900; 10,000 and 20,000 reads/sample for study A,B and C respectively).

## Tetanus specific IgG

Tetanus specific IgG in serum was tested in a pre-coated ELISA kit (XpressBio, mouse anti-tetanus toxoid IgG ELISA Assay, # IM-202, Thurmont, MD, USA). The standards, as well as all samples, were tested in duplicates on 96 wells plates. The plates were read at 405 and 450 nm on a microplate reader (Bio-Rad, model 550, Hercules, CA, USA), before being analyzed (Microplate manager, version 5.2.1, Bio-Rad). The samples were diluted 1:350 and treated according to manufacturer's instructions including the setup of a standard curve to calculate antibody titers from OD values.

## Serum cytokines

In serum diluted 1:1000 collected from the total blood collection at euthanasia from mice in study A cytokines Granulocyte Macrophage Colony-Stimulating Factor (GM-CSF), interferon gamma (IFN-γ), interleukin 1 alpha (IL-1α), IL-2, IL-4, IL-5, IL-6, IL-10, IL-17 and tumor necrosis factor (TNFα) were measured with a 10plex Th1/Th2 mouse FlowCytomix Kit (Bender MedSystems, Austria), as well as with FlowCytomix Simplex Kit for IL-18 and IL-12p70 (Bender MedSystems, Austria) on a BD FacsCanto Flow Cytometer (BD Biosciences, Denmark) in accordance with manufacturer's instructions. Data were processed with FlowCytomix Pro 2.4 Software (BD Biosciences).

## Spleen gene expression

Immediately after euthanasia the spleens of all mice from study B were harvested and frozen at -80 in RNAlater (Sigma Aldrich, St. Louis, USA). The extraction of total RNA and quality analysis, cDNA synthesis, specific primer design, pre-amplification and exonuclease treatment, and qPCR was performed as previously described [49]. Expressions of the genes in S1 Table were measured on a Fluidigm Biomark platform (Fluidigm Europe B.V., Amsterdam, Netherlands).

## Fluorescence-activated cell sorting (FACS)

Immediately after euthanasia, the spleens and mesenteric lymph nodes of mice in study C were sampled and cells were prepared in PBS according to our flow preparation protocol (https://osf.io/s47np/). The cells were analysed by an Accuri C6 flow cytometer and software (Accuri Cytometers Inc., Ann Arbor, USA) for the markers shown in Table 1 as previously described [44] using AH Diagnostics antibodies (Tilst, Denmark). For FACS 1–2 million cells were labelled. At each level 10,000 cells were counted. For rationalization some antibodies were only tested on randomly selected mice, and 5, 10 or 15 mice were tested on each level (Fig 5A and 5B). In some groups some of the preparations were unsuccessful and, therefore, n was reduced accordingly.

## Statistics

Antibody titers were ranked and analysed in Minitab 20.4 (Coventry, UK) by a one-way ANOVA with Tukey's post hoc comparisons (Study A and B) or a general linear model two-

**Table 1. Markers and gates for flow cytometry.** List of cell subsets investigated with fluorescence-activated cell sorting (FACS), as well as the marker used and gate from which it is defined, on cells extracted from mesenteric lymph nodes and spleen, from female BALB/c mice euthanized one week after a single injection of a tetanus vaccine.

| Cell line | Marker | Cell gate |
|---|---|---|
| T helper cells and regulatory T cells (Th & $T_{reg}$) | $CD4^+$ | All live cells |
| Activated $CD4^+$ T-cells | $CD25^+/ CD69^+ / CD25^+CD69^+$ | $CD4^+$ |
| Regulatory T cells ($T_{reg}$) | $FoxP3^+$ | $CD4^+ CD3^+$ |
| Follicular T-cells | $CXCR5^+$ | $CD4^+ CD3^+$ |
| Cytotoxic T-cells | $CD8^+$ | All live cells |
| Activated cytotoxic T-cells | $CD69^+$ | $CD8^+$ |
| B-cells | $CD19^+$ | All live cells |
| Macrophages | $F4.80^+$ | All live cells |
| Activated DCs | $CD80^+$ | $CD11c^+$ |
| Tolerogenic DCs | $CD11b^+$ | $CD11c^+$ |
| Major Histocompatibility complex II | $MHCII^+$ | $CD11c^+$ |

way ANOVA (Study C). Variances of antibody titers were compared with Levene's test (Minitab). ELISA data were corrected for plate variation prior to statistical analysis. FACS data were ranked and tested by a two-way general linear model for the factors gluten-free and antibiotics (Minitab) and Dunnett's post hoc test compare the gluten-free, antibiotics and combined groups with the gluten group (GraphPad Prism 9, San Diego, USA). The F test was applied to compare the variation in titers between two groups (Prism). Weight data were tested in a two-way general linear model for the factors gluten-free (Study A), antibiotics (Study B), and vaccination (Study A), and with a fixed nesting of time (Minitab). Fold change data from gene expression analysis were subjected to Kruskal–Wallis test (Minitab) and P-values were corrected with false discovery rate (FDR) with the Benjamini Krieger Yekutieli method (Prism). Growth curves were compared by a repeated measures ANOVA (Prism). Significance level was set at $P < 0.05$.

The compare_alpha_diversity workflow (QIIME 1.9.1) was used to test differences in alpha diversity indices between categories (nonparametric t-test with 999 Monte Carlo permutations). The compare_categories workflow (QIIME 1.9.1) was used to perform Permutational analysis of variance (PERMANOVA) on the distance matrices to test for dissimilarities between groups. Differences in the taxa relative distribution between categories were tested using analysis of composition of microbes (ANCOM) [50]. The observation_metadata_correlation workflow (QIIME 1.9.1) was used to find Pearson correlations between specific operational taxonomic units (OTUs) and tetanus vaccine specific immunoglobulin (Ig) G level with subsequent FDR correction (Q value) in Study A, B and C. Correlations were conducted using the QIIME script observation_metadata_correlation.py. The Pearson test was chosen to calculate the correlation between the IgG values and the rarefied OTU tables, 1000 permutations were applied for calculating bootstrapped p-values.

## Results

### Gluten-free diet as well as treatment with ampicillin had significant impact on the gut microbiota compared to mice fed standard diet

We sequenced the GM to describe the impact of a gluten-free diet or antibiotics on the GM. Mice fed the gluten-free Altromin modified diet for eight weeks in study A developed a GM distinct from the standard Altromin diet fed mice (Fig 2A and 2B, P = 0.01) with the gluten-free diet induced GM being characterized by an increased relative abundance of *Parasuterella*

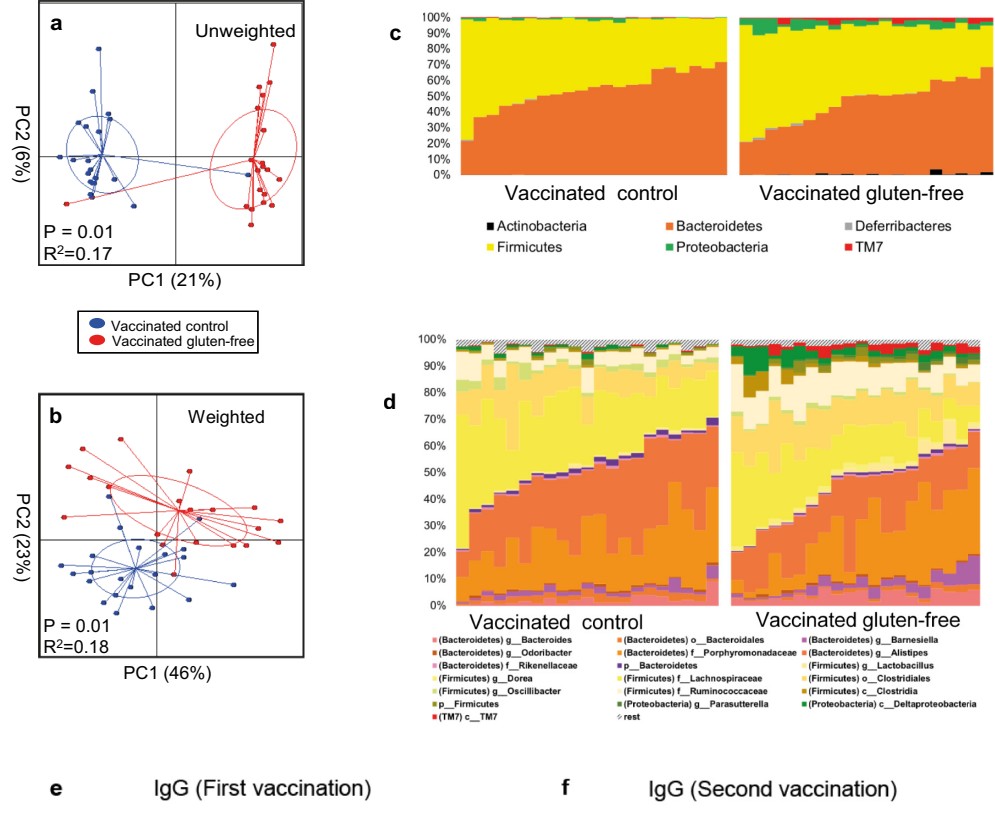

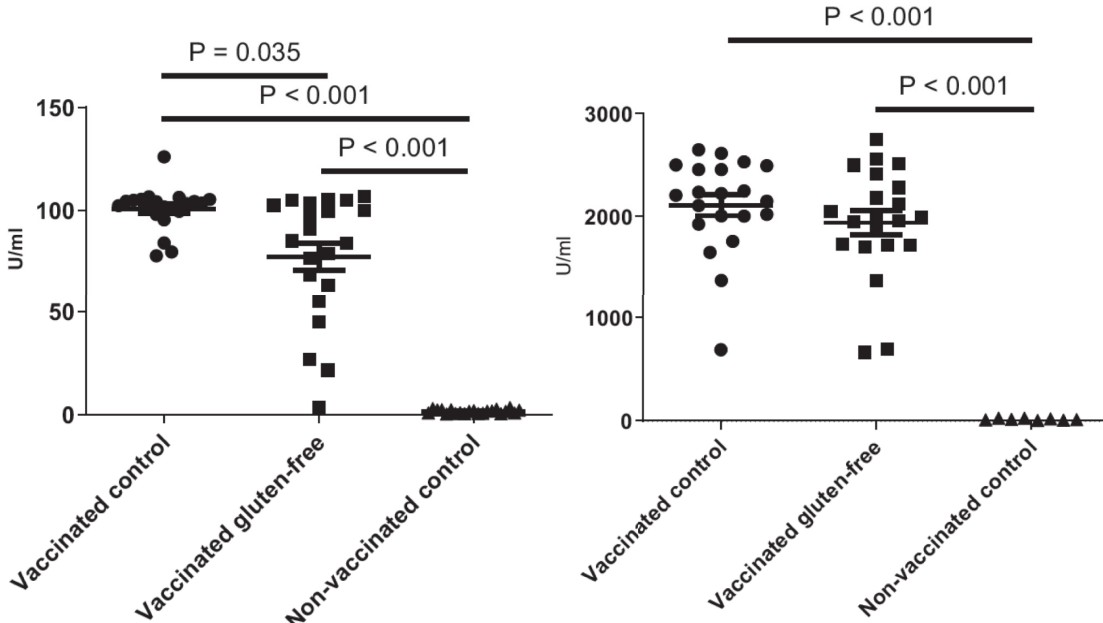

**Fig 2. The impact of a gluten-free diet on gut microbiota and anti-tetanus IgG after tetanus vaccination.** BALB/cBomTac mice were vaccinated or not and fed a standard Altromin wheat based diet ('gluten'), or they were vaccinated and fed a modified Altromin diet, in which wheat protein was replaced with casein ('gluten-free'). Unweighted (a) and weighted (b) PCoA plots with Permutational multivariate analysis of variance (PERMANOVA) results, relative abundance bar plots on phylum (c) and OTU (d) level after microbiota characterization on feces sampled three and a half week after second vaccination, and anti-tetanus IgG (median) two and a half week after first (e) and three and a half week after second (f) tetanus vaccination.

**Table 2. Normalized relative abundances (Mean ± s.d.) of gut microbiota bacteria differing significantly (ANCOM) between female BALB/cBomTac, which were either fed standard wheat based Altromin diet (Gluten) or a modified gluten-free Altromin diet (Gluten-free) from the age of six to fourteen weeks.** All mice were vaccinated with a tetanus vaccine. FDR-corrected q-value was 0.010.

| Taxonomy | Gluten | Gluten-free |
|---|---|---|
| | n = 21 | n = 20 |
| Bacteria; Bacteroidetes; Bacteroidia; Bacteroidales; Prevotellaceae; Other; Other | 23.1 ± 12.4 | 0.8 ± 2.6 |
| Bacteria; Proteobacteria; Betaproteobacteria; Burkholderiales; Alcaligenaceae; *Parasutterella*; Other | 1.8 ± 2.9 | 47.9 ± 42.4 |
| Bacteria; Firmicutes; Clostridia; Clostridiales; Lachnospiraceae; *Dorea*; Other | 13.6 ± 7.6 | 81.9 ± 41.6 |
| Bacteria; Firmicutes; Clostridia; Clostridiales; Ruminococcaceae; *Ruminococcus*; Other | 8.4 ± 9.7 | 0.2 ± 0.5 |
| Bacteria; TM7; TM7_genera_incertae_sedis; Other; Other; Other; Other* | 12.8 ± 9.9 | 119.0 ± 74.0 |
| Bacteria; Actinobacteria; Actinobacteria; Bifidobacteriales; Bifidobacteriaceae; *Bifidobacterium*; Other | 0 ± 0 | 25.4 ± 39.1 |

* Reclassified as Saccharibacteria_ Saccharimonadaceae.

spp., Lachnospiraceae, TM7 and *Bifidobacterium* spp. and reduced relative abundances of *Ruminococcus* spp. and Prevotellaceae (Fig 2C and 2D and Table 2). Ampicillin was an effective modulator of the GM composition in study B (Fig 3A and 3B, P = 0.01). The influence on distribution of microbes was already obvious on the phylum level, as the antibiotic treatment eliminated the Bacteroidetes in many of the animals and raised the relative abundance of Proteobacteria, such as Pseudomonadaceae and Rhizobiaceae as well as *Cyanobacteria* (Fig 3C and 3D). Interestingly, we only observed group differences in beta-diversity but not alpha-diversity. No GM differences were found between vaccinated and unvaccinated mice on the same standard Altromin diet. Vaccine boosting did not significantly alter the GM. The effect of the gluten-free diet on the GM in study C was too small to be detected in the model due to the major effect of the antibiotics (P = 0.00) (Fig 4A–4D). Mice fed the Altromin modified gluten-free diet weighed significantly more than mice fed the standard Altromin diet in study A (S1A Fig, P = 0.000). Antibiotics in the drinking water reduced the weight significantly in study B (S1B Fig, P = 0.001), but this was mostly due to a weight decrease in the first weeks on antibiotics. We found a significant positive correlation on genus level after FDR correction of especially species within the phylum Bacteroidetes with the pooled IgG levels of both time points in the antibiotic treated mice compared to their controls (Study B), while no correlations were found after the first vaccination, in single groups or in study A (S3 Table).

## Gluten-free diet and antibiotics lowered antibody production after the first vaccination

To describe the impact of a gluten-free diet or antibiotics we monitored serum antibodies to tetanus after the first and second vaccination. Diet had a significant impact on tetanus specific IgG titers monitored two weeks after first vaccination (study A), as mice fed the Altromin modified diet without gluten had both lower titers (P = 0.005) and higher variation (P = 0.000) (77.4 U/ml ± 30.4) compared to the vaccinated control group (100.7 U/ml ± 10.2) (Fig 2E). All titers were ten fold higher three weeks after boosting, but at this time point there no longer were significant differences between the titers or their variation (Altromin modified: 1942.0 U/ml ± 539.2; Altromin: 2110.0 U/ml ± 462.1) (Fig 2F). The non-vaccinated mice after both first and second placebo injection had titers close to zero (Fig 2E and 2F). In Study B, in which all mice were vaccinated, tetanus specific IgG titers monitored three weeks after the first vaccination were significantly lower in the ampicillin treated group (161.6 U/ml ± 40.0) (P = 0.011) compared to the untreated control mice (203.0 U/ml ± 63.5), and the ampicillin treated mice had a borderline lower variation in the titer as shown by Levene's test (P = 0.080) (Fig 3E).

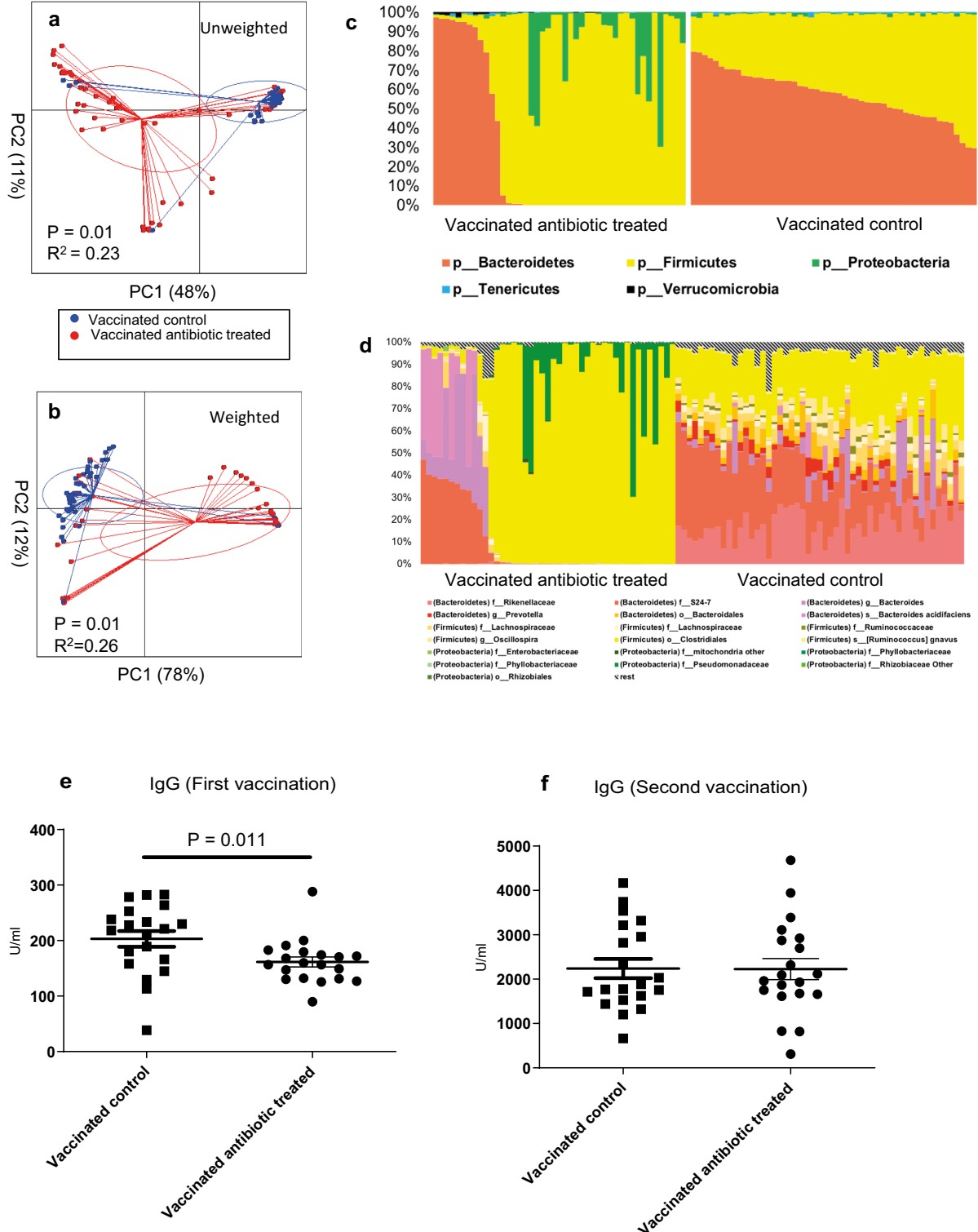

**Fig 3. The impact of a antibiotics on gut microbiota and IgG response to tetanus vaccination.** BALB/cBomTac mice were vaccinated twice with a tetanus vaccine, either in combination with antibiotics (ampicillin) in the drinking water (1g/L) or with pure drinking water. Unweighted (a) and weighted (b) PCoA plots with PERMANOVA results, relative abundance bar plots on phylum (c) and OTU (d) level after microbiota characterization on feces sampled three weeks after second vaccination, and anti-tetanus IgG (median) two weeks after first (e) and three weeks after second (f) tetanus vaccination.

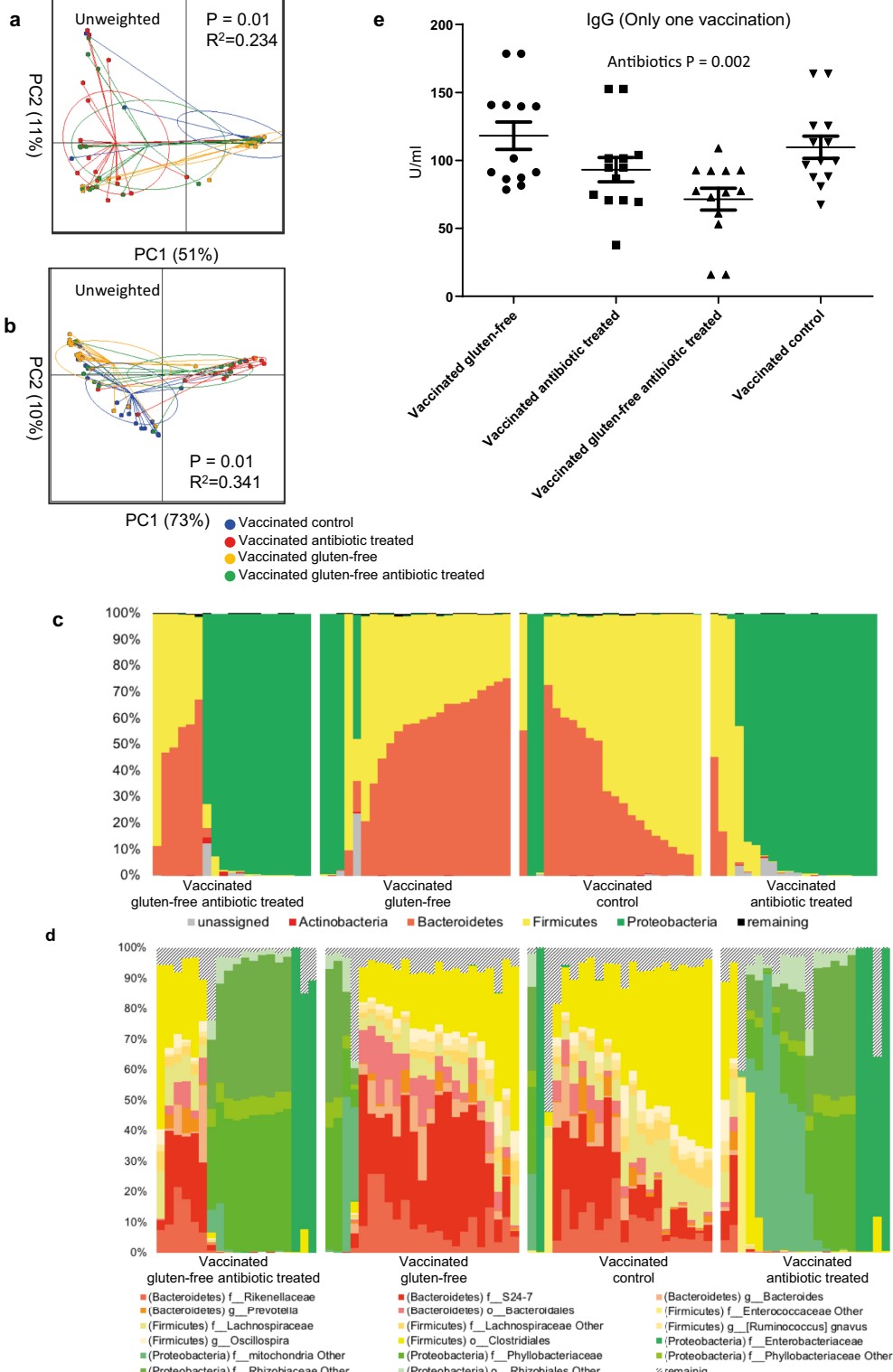

**Fig 4. The impact of a gluten-free diet and antibiotics on gut microbiota and IgG response to tetanus vaccination.** BALB/cBomTac mice were either fed a standard Altromin wheat based diet ('gluten'), or they were fed a modified Altromin diet, in which wheat protein was replaced with casein ('gluten-free'). Mice on both diets were either given ampicillin in the drinking water (Antibiotics; gluten-free + antibiotics) or given pure drinking water (Gluten-free; vaccinated control). All mice were vaccinated with a tetanus vaccine and one week later blood was sampled. Unweighted

(a) and weighted (b) PCoA plots with PERMANOVA results, relative abundance bar plots on phylum (c) and OTU (d) level after microbiota characterization on feces sampled at vaccination, and anti-tetanus IgG (median) one week after tetanus vaccination.

This difference had vanished two weeks after boosting (Antibiotics: 2229.0 U/ml ± 1061.0; Vaccinated control: 2241.0 U/ml ± 964.8) (Fig 3F). The tetanus specific IgG titers monitored at euthanasia in the combined gluten-free and antibiotics treated mice in study C one week after vaccination also showed a lowering effect of antibiotics (P = 0.008, Fig 4E), while there at this early time point was no effect of the gluten-free diet.

## The fractions of essential immune cell subsets and their activation were reduced in the mesenteric lymph nodes, while it was increased in the spleens of mice fed the gluten-free diet

To describe the impact of a gluten-free diet and/or antibiotics on the immediate immune cell activation after a single tetanus vaccination we counted immune cells one week after the vaccination (Study C). Feeding the Altromin modified gluten-free diet reduced the overall fraction of T cells (CD3$^+$) (P = 0.000), and the more specific fractions of cytotoxic T cells (CD8$^+$: P = 0.002), and T$_{reg}$ (FoxP3$^+$CD3$^+$CD4$^+$: P = 0.013) in the mesenteric lymph nodes (MLN) (Fig 5A). It also reduced the activation of both CD4$^+$ (CD25$^+$: P = 0.000; CD69$^+$: P = 0.000; CD25$^+$/CD69$^+$: P = 0.000) (Fig 5A), and CD8$^+$ (CD69$^+$: P = 0.000) (Fig 5A) T cells in the MLN. In the spleen the Altromin modified gluten-free diet reduced the fraction of cytotoxic T cells (CD8$^+$: P = 0.007 (Fig 5B). However, in contrast to the MLN the fraction of T$_{reg}$ (FoxP3$^+$CD4$^+$CD3$^+$: P = 0.013), as well as the fractions of activated CD4$^+$ T cells (CD25$^+$CD4$^+$: P = 0.000; CD69$^+$CD4$^+$: P = 0.000; CD25$^+$CD69$^+$CD4$^+$: P = 0.000), and activated CD8$^+$ cytotoxic T cells (CD69$^+$CD8$^+$: P = 0.000) were increased by the gluten-free diet (Fig 5B). Furthermore, the gluten-free diet increased the fraction of dendritic cells (CD11c$^+$: P = 0.007), i.e. both activated dendritic cells (CD80$^+$CD11c$^+$ P = 0.001), and tolerogenic dendritic cells (CD11b$^+$CD11c$^+$: P = 0.000) were increased by the gluten-free diet (Fig 5B).

Antibiotics significantly reduced the fraction of T cells (CD3$^+$; P = 0.006) and CD4$^+$ T cells (P = 0.003), and borderline reduced the fraction of B cells (CD19$^+$: P = 0.086) in the MLN (Fig 5A). However, there was no specific effect on the fractions of CD4$^+$CD3$^+$ cells (T$_{reg}$ and Th) or FoxP3$^+$CD4$^+$CD3$^+$ cells (T$_{reg}$). In the spleen antibiotics lowered the fraction of activated T cells (CD69$^+$/CD4+: P = 0.040; CD69$^+$/CD8+: P = 0.003) and the fraction of tolerogenic dendritic cells (CD11c$^+$/CD11b$^+$: P = 0.037) (Fig 5B).

## Diet or antibiotics had only minor impact on systemic cytokine levels or spleen gene expression

To describe whether the gluten-free diet or the vaccination had an impact on serum cytokine levels the cytokines IL-1α, IL-2, IL-5, IL-6 and GM-CSF were monitored at the termination of study A. They were all present in detectable amounts in blood serum, but there were no significant differences in relation to either diet or vaccination (data can be found in the repository).

To detect whether antibiotics or vaccination had an impact on immune gene expression the spleens were sampled for gene expression by qPCR at the termination of study B. Although there was a significant difference in fold changes from the spleens of the genes *Cxcl10*, *Hp*, *Ifng*, *Klf2* and *Tlr4* in individual Kruskal Wallis tests, no significant differences in gene expression were revealed after false discovery rate (FDR) correction (S2 Table).

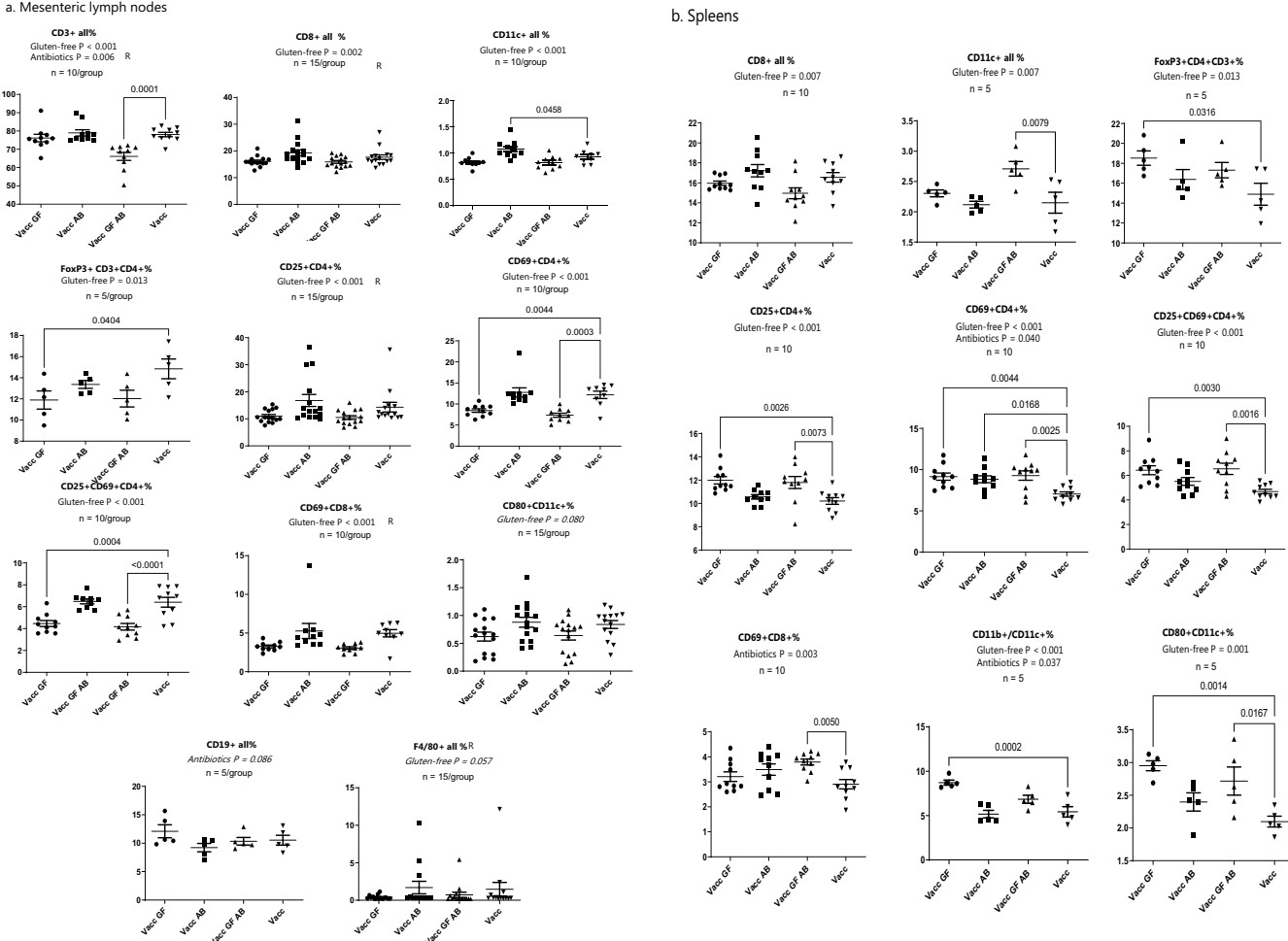

**Fig 5. Effects of a gluten-free diet or antibiotics on immune cell counts as a response to a tetanus vaccine.** Flow assisted cell sorting (FACS) data (See Table 1) from the mesenteric lymph nodes (A) and the spleens (B) of female BALB/cBomTac, which were either fed standard wheat based Altromin diet or a modified gluten-free Altromin diet (GF). Mice on both diets were either given ampicillin (AB) in the drinking water (Antibiotics; gluten-free + antibiotics) or given pure drinking water (Gluten-free; vaccinated control). All mice were vaccinated with a tetanus vaccine (Vacc) and one week later MLN were sampled. Only data being significantly different between the groups are shown. R indicates that data have been ranked before statistical tests due to the lack of equal variances and/or normal distribution. Over all p-values are the outcome of two-way ANOVA for the two factors 'gluten-free' and 'antibiotics', while the values on the comparison lines are the p-values for the post-hoc Dunnett's test comparing the Vacc GF, Vacc AB and the Vacc GF AB with the Vacc group.

## Discussion

Feeding a gluten-free diet to mice altered the GM, and it lowered the initial anti-tetanus IgG response to the potent tetanus toxoid vaccine with the strong adjuvant aluminium-oxide hydrate. As expected antibiotics also induced GM changes and reduced anti-tetanus IgG following the first tetanus vaccination. There seemed to be a clear difference between the way that a gluten-free diet and antibiotics had an impact on antibody response. Antibiotics did not in the increase the variation in response, and except for one mouse all antibiotics treated mice had a response below the average for non-treated mice. After boosting the responses in both gluten-free and antibiotic fed mice were comparable to the control mice.

In humans, it has been shown that gluten-free diet induced GM changes are essential for the increased T$_{reg}$ levels [51], although also a direct anti-inflammatory effect of a gluten-free diet has previously been shown in mice [52]. From studies in two-generation studies in non-

obese diabetic mice, it is known that the effects of a gluten-free diet on the immune system can be transferred over generations both dependently and independently of the GM [15, 53]. In the short term study C it was difficult to verify any major anti-tetanus IgG titer effect of the gluten-free diet one week after vaccination, and the impact of the gluten-free feeding was also minor compared to the two other studies. However, there was still in terms of cell counts a clear impact on the immune system, which therefore either responds to even small GM changes and/or directly to the changed diet composition. Whether the impact on vaccine response is GM dependent might be further elucidated through a fecal matter transplantation to germ-free mice.

It might be important for future mechanistic studies that there was an increased relative abundance of *Bifidobacterium* spp. in mice fed the gluten-free diet. This is in accordance with our previous observations that the gluten-free diet increases the number of $T_{reg}$ [26], which fits well with a reduced vaccine response. In the present study we found that the gluten-free diet lowered the levels of $T_{reg}$ locally in the gut, while in the spleen, which should be regarded as an expression of the systemic effect, the gluten-free diet increased $T_{reg}$ fractions, $CD4^+$ T cell activation, and tolerogenic dendritic cell fractions and activation, thereby extending the downregulating effect of the $T_{reg}$ [54]. Therefore, the systemic effect of the gluten-free diet seems mainly tolerogenic. The inter-individual variation in *Bifidobacterium* spp. relative abundances of the gluten-free fed mice was huge, while *Bifidobacterium* spp. were completely absent in all gluten fed mice. This may very well explain the huge inter-individual variation in the antibody response of the gluten-free fed mice. Feeding oligosaccharides, which are used to promote anti-inflammatory bacteria, also negatively affected IgG titers to a tetanus toxoid vaccine in male mice [55]. We observed a correlation with anti-tetanus IgG in study B, i.e. the antibiotics treated mice. However, this should not be over-interpreted, because in the antibiotics treated group, which had a reduced anti-tetanus IgG production, there was reduced relative abundances of Bacteroidetes, and we found no correlations in study A.

In accordance with our previous observations both a gluten-free diet and antibiotics increased the relative abundance of *Proteobacteria*, such as *Parasutterella* spp. [15]. These contain the MAMP lipopolysaccharide (LPS), which through TLR4 stimulates TNF-α production [56]. We did observe decreased *Tlr4* expression in antibiotics treated mice, although non-significant after FDR correction. LPS from the gut drives expansion of $T_{reg}$ and reduces later life severity of ulcerative colitis in mice [57, 58], and we have previously observed that in both Peyer's patches and in the spleen a low concentration of dietary LPS induces the formation of $CD103^+$ dendritic cells [59], which are required for the induction of tolerogenic responses [60]. A low amount of proteobacterial LPS in the gut the first 1–2 years of life in Finnish compared to Russian children has been used as an explanation of their lower number of $T_{reg}$ and their increased risk of autoimmune diseases [61]. Such differences between children in different geographical areas are also likely to explain vaccine response differences. In horses a relative abundance of Prevotellaceae disposes for a high expression of the $T_{reg}$ marker FoxP3 [62], while a high relative abundance disposes for glucose intolerance in mice [63]. Immediately after treatment ampicillin has been shown to reduce Peyer's patch $T_{reg}$ fractions [31]. Although antibiotics seemed to have a weaker effect than the gluten-free diet in the spleen, it downregulated tolerogenic dendritic cell activation, and it increased cytotoxic T cell activation. Therefore, both locally in the MLN and systemically in the spleen, antibiotics and gluten-free diet seem to follow different pathways in downregulating IgG. Ampicillin enters the blood stream and may elicit systemic effects by itself, while a gluten-free diet mostly has its effect on the GALT, and its systemic effects are probably secondary.

In this study, mice were treated with ampicillin throughout the study, which is much longer than humans normally would be treated. During ampicillin treatment the majority of naturally gut inhabiting bacteria will either be eradicated or heavily suppressed, and when ampicillin

treatment in mice is terminated, some microorganisms will recover from the host, while other will recover from the environment, and, therefore, ampicillin treated mice will have a microbiota different from non-treated mice [31]. It is unknown whether such a microbiota change will have long term impact on the immune system, and, therefore, our studies cannot be used to state whether the general use of antibiotics in children hampers their responses to vaccines; especially not if they are not treated during vaccination.

The dietary impact on the microbiome not only relates to the bacterial community of the host. Even specific pathogen free mice harbours a gut virome both in the form of phages [64] and mammal viruses [65]. These may have a major impact on host responses [66]. The microbiomic as well as the host related digestion produces a metabolome, and the components of this may equally well facilitate differences in immune responses [67]. From an applied point of view it may be easier to transplant phages than entire microbiotas to change the microbiota in the direction of a stronger vaccine response [68].

Laboratory mice are often very clean with a low microbiota diversity [69], which may be regarded as unfavourable in vaccine development using mice [70]. Human transcriptional responses to influenza vaccination are better recapitulated and humoral responses are dampened in 'dirty' mice [71]. The increased Proteobacteria dominance with increased LPS impact in both gluten-free fed and antibiotic treated mice, may, although contradictory, make them more 'dirty' comparable to the situation of some human populations. Additionally, our observations add to the understanding that mouse studies are often difficult to reproduce. In large rodent facilities standard diets are purchased in bulk and all mice in different studies are currently fed the latest and often different batches during the same study, although batch variation for certain micronutrients may be substantial in natural ingredient diets [72]. Therefore, scientists should be more aware of reporting exact data on diet use in scientific papers, for example by adhering to the ARRIVE guidelines [73].

## Limitations

Studies in mice typically give some basic information, but they are quite often not translatable to humans, i.e. our data cannot be taken as a solid statement that a gluten-free diet has the same impact in humans. Also, there were differences in the experimental setup when testing the impact of a gluten-free diet and the impact of antibiotics. So, although it seemed as if the way these two factors impacted the vaccine responses differed, this interpretation should be made with the precaution that the two studies are not directly comparable. The two-way setup for testing the impact on cell counts does not include a non-vaccinated group, so it is only possible to make conclusions on the relative impact of the experimental factors. Finally, these studies were performed with a tetanus toxoid vaccine, which globally in humans is known as a highly efficient vaccine [39], and to which the immune system responds in ways, which can be different from responses to other vaccines. The effectiveness of viral vaccines seems to vary far more. Titers to influenza varies significantly between vaccinated humans in different geographical regions [1], and even the most efficient COVID-19 vaccines do not offer full protection against infection in humans [74]. Therefore, our studies in mice should not be used to question the efficiency of tetanus vaccination in humans, and from a more applied point of view it would be equally or more relevant to study the impact of the GM on such vaccines; especially in the light of the COVID-19 pandemic.

## Conclusions

We studied the impact of the GM on the response of a tetanus vaccine. From a basic point of view our experiment supports the hypothesis that GM has an impact on the response to

vaccines [75]. The GM with its potential to engage both innate and adaptive immune responses [76, 77], is, therefore an obvious target for manipulation in regard to optimizing vaccine strategies. Diet is a strong modulator of the GM [13], and therefore diet and indirectly factors such as geographical habitat need consideration in the development of vaccines in studies in mice as well as in humans.

## Supporting information

**S1 Fig.** Growth curves for mice fed a gluten free diet (a) or antibiotics (b). (a) BALB/cBomTac mice were vaccinated with a tetanus vaccine (Vaccinated control) or not (Control) and fed a standard Altromin wheat based diet ('gluten'), or they were vaccinated and fed a modified Altromin diet, in which wheat protein was replaced with casein ('gluten free'). (b) BALB/cBomTac mice were vaccinated twice with a tetanus vaccine, either in combination with ampicillin (antibiotics) in the drinking water or with pure drinking water (control). Borderline p-values are written in italics.
(PDF)

**S1 Table. Genes for which expressions were monitored by qPCR in the spleens of tetanus vaccinated antibiotics treated and control mice.**
(PDF)

**S2 Table. Spleen gene expressions for which a significant difference between tetanus vaccinated antibiotics treated and control mice was found by Kruskal–Wallis test (P).** No significances were found if P-values were corrected by false discovery rate (Q).
(PDF)

**S3 Table. Correlations of taxa with anti-tetanus IgG in BALB/cBomTac mice vaccinated once or twice with a tetanus toxoid vaccine.** Only correlations significant after FDR correction are shown.
(DOCX)

## Acknowledgments

Mette Nelander and Helene Farlov are kindly acknowledged for taking care of the animals.

## Author Contributions

**Conceptualization:** Pernille Kihl, Karsten Buschard, Dennis S. Nielsen, Axel Kornerup Hansen.

**Data curation:** Lukasz Krych.

**Formal analysis:** Axel Kornerup Hansen.

**Funding acquisition:** Axel Kornerup Hansen.

**Methodology:** Pernille Kihl, Lukasz Krych, Ling Deng, Lars H. Hansen, Søren Skov, Dennis S. Nielsen.

**Project administration:** Pernille Kihl, Axel Kornerup Hansen.

**Writing – original draft:** Pernille Kihl, Lukasz Krych, Ling Deng, Karsten Buschard, Axel Kornerup Hansen.

**Writing – review & editing:** Pernille Kihl, Lukasz Krych, Ling Deng, Lars H. Hansen, Karsten Buschard, Søren Skov, Dennis S. Nielsen, Axel Kornerup Hansen.

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
