## [Decision Letter · Decision Letter 0]

27 Jun 2021

PONE-D-21-12086

Dietary Gut Microbiota Perturbations Influence Murine Vaccine ResponseDietary Gut Microbiota Perturbations Influence Murine Vaccine Response

PLOS ONE

Dear Dr. Kornerup Hansen,

Thank you for submitting your manuscript to PLOS ONE. After careful consideration, we feel that it has merit but does not fully meet PLOS ONE’s publication criteria as it currently stands. Therefore, we invite you to submit a revised version of the manuscript that addresses the points raised during the review process.

We look forward to receiving your revised manuscript.

Kind regards,

Anne Wertheimer, PhD

Academic Editor

PLOS ONE

Journal Requirements:

4. Please include your tables as part of your main manuscript and remove the individual files. Please note that supplementary tables (should remain/ be uploaded) as separate "supporting information" files.

5. Please upload a copy of Supporting Information Table 1 and 2 which you refer to in your text on page 9 and 14.

6. Thank you for stating the following in the Financial Disclosure section:

[The study was supported by a grant (2013-4) from LIFEPHARM (www.lifepharm.ku.dk) to AKH. The funder had no role in the project.]. 

We note that you received funding from a commercial source: Lifepharm

Additional Editor Comments:

Hansen et. al present studies which address important aspects relevant to vaccination success. Examination of diet as well as antibiotic therapy are both relevant conditions. There are a few areas of concern which must be appropriately addressed prior to acceptance.

Major concerns:

Figure 2e and Figure 4e though providing the same type of data (IgG response after one vaccination) yet in the different study groups (Study A and C) the vaccinated control and vaccinated gluten Free cohort had statistically different levels of IgG in Study A but not in Study C. The variability of the vaccinated control cohort was also considerably less in Study A. Please address.

Since there was neither a gluten-free nor an antibiotic treatment cohort which were unvaccinated all reference to the gluten free cohort and antibiotic treated cohort must be stated as the "vaccinated" gluten free cohort and the "vaccinated" antibiotic treated cohort for clarity. Sentences such as line 298 "To describe whether the gluten-free diet or the vaccination had an impact on serum..." as well as line 302 "To detect whether antibiotics or vaccination had an impact on immune gene expression" must be restated and analysis adjusted accordingly because there was not a antibiotic cohort that was not vaccinated. For example line 302 could be rephrased to state "To detect whether vaccination in the presence of antibiotics had an impact..."

Verify that the n for each figure is correctly reflected in the number of symbols for each cohort. For example in 4e there appear to be an n=13 in the figure yet in the text the cohort is n=15. Define and explain the number of animals represented in figures 5a and 5b. It appears that the number of animals used in this series of experiments varies and is not explained. Why for example were only 5 animals used in the FoxP3+CD3+CD4 vs the CD25+CD4 panel. In addition lines illustrated where the p value applies must be added.

Provide additional detail on the sample preparation and analysis for flow cytometry either in the methods section or within Table 1; specifically the number of cells stained and the final number of cells within the analysis gate.

For p=0.000 restate as p< 0.005. The images were very pixelated.

The additional findings from Reviewer 1 must also be addressed in your resubmission.

Reviewers' comments:

Reviewer's Responses to Questions

**Comments to the Author**

1. Is the manuscript technically sound, and do the data support the conclusions?

Reviewer #1: Partly

2. Has the statistical analysis been performed appropriately and rigorously? 

Reviewer #1: Yes

3. Have the authors made all data underlying the findings in their manuscript fully available?

Reviewer #1: No

4. Is the manuscript presented in an intelligible fashion and written in standard English?

Reviewer #1: Yes

5. Review Comments to the Author

Reviewer #1: In this manuscript Kihl and colleagues have looked at the effect of diet and vaccine response. While this manuscript focusses on the effect of gluten-free diet on many avenues such as gut microbiome, host immune response, tetanus specific IgA and serum cytokines, some questions still remain.

Major concerns:

1. It seems like the mouse experiments used only female BALB/cBomTac mice for their experiments. Please explain why male mice were not used for these experiments?

2. Study A: Impact of gluten on primary and boosting vaccination cohort. While there were two groups fed normal diets (vaccinated and placebo), there is no placebo group for the gluten free diet (Altromin modified). These analyses, therefore do not allow the separation of the effects of gluten-free diet from that of vaccinations alone.

3. It is not clear why samples were collected the same day as vaccination for Study A and C and a few days after vaccination in Study B.

4. Was ampicillin administered throughout the duration of the study? Did these animals get diarrhea? If yes, then did the animals that developed diarrhea show differential gut microbiota and immune response as well as serum cytokines etc.

Minor concerns:

1. The figures are pixelated and very hard to read.

6. PLOS authors have the option to publish the peer review history of their article (what does this mean?). If published, this will include your full peer review and any attached files.

Reviewer #1: No

---

## [Author Response · Author response to Decision Letter 0]

26 Aug 2021

Dear editor

I hereby submit the revised manuscript.

We have at our best tried to follow the valuable comments of ypu and the reviewer, as described in the cover letter.

Best wishes

Axel Kornerup Hansen

---

## [Decision Letter · Decision Letter 1]

13 Jan 2022

PONE-D-21-12086R1Dietary Gut Microbiota Perturbations Influence Murine Vaccine ResponsePLOS ONE

Dear Dr. Kornerup Hansen,

Thank you for submitting your manuscript to PLOS ONE. After careful consideration, we feel that it has merit but does not fully meet PLOS ONE’s publication criteria as it currently stands. Therefore, we invite you to submit a revised version of the manuscript that addresses the points raised during the review process.

 Both reviewers note a substantial number of serious concerns that must be addressed before further consideration can be made.

We look forward to receiving your revised manuscript.

Kind regards,

Brenda A Wilson, Ph.D.

Academic Editor

PLOS ONE

Additional Editor Comments (if provided):

As the second academic editor to review this manuscript, it was difficult to discern where the improvements in the revision were since the second set of reviewers expressed very similar concerns to that of the first round of reviewers as well as additional ones. After reading the manuscript through myself, I would agree with these two reviewers that major revision of the manuscript is needed before further consideration can be made.

Reviewers' comments:

Reviewer's Responses to Questions

**Comments to the Author**

1. If the authors have adequately addressed your comments raised in a previous round of review and you feel that this manuscript is now acceptable for publication, you may indicate that here to bypass the “Comments to the Author” section, enter your conflict of interest statement in the “Confidential to Editor” section, and submit your "Accept" recommendation.

Reviewer #2: (No Response)

Reviewer #3: (No Response)

2. Is the manuscript technically sound, and do the data support the conclusions?

Reviewer #2: Partly

Reviewer #3: Yes

3. Has the statistical analysis been performed appropriately and rigorously? 

Reviewer #2: Yes

Reviewer #3: No

4. Have the authors made all data underlying the findings in their manuscript fully available?

Reviewer #2: Yes

Reviewer #3: Yes

5. Is the manuscript presented in an intelligible fashion and written in standard English?

Reviewer #2: No

Reviewer #3: No

6. Review Comments to the Author

Reviewer #2: The paper describes an observation that mice when fed with a gluten-free diet or treated with ampicillin have slightly lower primary antibody IgG response against a tetanus vaccine, but the secondary Ab response seems normal. The authors conclude that dietary or gut microbiota factors should be considered when testing vaccines. The data in most part are correlative, no mechanistic study is engaged.

(1) Fig5 is problematic, it is not known which two groups are different from each other simply by showing a p-value. The same thing is true for Fig4a,b, and e.

(2) Ampicillin treatment in study B is a little bit too long. Does short-term one-week treatment also have an impact?

(3) Can the author demonstrate that the gluten-free diet-induced microbiota changes drive the lower primary antibody response by fecal transfer type of experiment? Or can the author demonstrate which cell population influenced by the diet reduced the antibody response?

(4) The language can be modified further. p10 L223, please delete the ' mark.

Reviewer #3: General comments

This is the reviewer’s first reading of this article, although it appears it has already gone through one round of reviews. Apologies to the authors if the have already gone through a round of reviews.

The article aims to characterize the impact of microbiome alteration (either through diet or antibiotics) on tetanus vaccine immunogenicity in mice. The article is a nice addition to the literature that suggests that alteration of the intestinal microbiome alters host immune response to both vaccines and pathogens. The authors have performed experiments with a large sample sizes and evaluated microbiome, immune cells, and gene expression resulting in an interesting data set that suggests gluten free-diet and antibiotic administration diminish anti-tetanus antibodies following intial vaccination, possibly through different mechanisms.

Major comments

The reviewer requests a better rationale for why tetanus vaccine was used for this study, given that it is known to elicit consistently high antibody responses across geography and across ages.

Also suggest to reference more of the recent (human and murine) literature on how microbiota alters vaccine immunogenicity.

Study design - Studies A, B and have differing intervals of treatment (antibiotic/diet) and differing timing of prime and boost of vaccination as well as measurement of antibody responses making it difficult to extrapolate results across the studies. The rationale for choices are not clearly described.

Analysis – Given the use of antibody titers, request that the authors repeat antibody statistical analyses using transformed data/geometric mean titers.

Article can often be confusing to read and suggest input from a native English speaker. Some hypotheses in introduction and conclusion suffer from large jumps across subjects without clear delineation of whether data comes from human or animal subjects. Suggest to review and adjust claims accordingly.

Minor comments

Abstract

Line 26 Wording is confusing, isn’t the primary purpose of the study to evaluate the effect of diet ( gluten-free or gluten-containing) and antibiotics (ampicillin or not) on tetanus vaccine immunogenicity? Suggest to adjust wording

Line 28 suggest change ‘lowered’ to ‘was associated with’

Line 29 low-responder not defined in abstract or text, suggest to define if this is a statistical outcome or otherwise remove

Line 29 relative abundance or abundance?

Line 30 remove word massively, change to significantly if this is the case. If hypothesis is that diet altered microbiota which then altered vaccine response, suggest to start with change in microbiota in this sentence

Line 30, suggest to remove ‘similarly’- suggests that antibiotics altered microbiota similarly to diet?

Line 31, add significance for all findings

Line 31, define ‘boosting’

Line 32, what was definition of” influenced’’?

Line 36, suggest to alter sentence, “these results suggest …..”

Introduction –

Line 44 The authors describe diminished vaccine efficacy in low-middle income settings, however tetanus vaccines have never been shown to have diminished efficacy in low or middle income settings. Rather, they have high immunogenicity and good protection globally, with possible higher immunogenicity in low-income settings. Please correct

Line 46 Suggest to use the term low-middle income and high income and not ‘developing”and developed”as terminology

Line 48 – all citations have to do with changing vaccine immunogenicity by genetics and age and do not support differential vaccine performance by geography. Please correct

Line 71, remove the word óbviously’, suggest to make less definitive statements, add “can cause” and add that this study was in mice

Line 74, modify that this is a study that was performed in mice

Line 76, add that this is for influenza vaccination

Line 83, cite literature that shows this

Line 88, citation is not about Bangladeshi children

Study Design

Line 136, what was the rationale for the different timing of tetanus vaccination between the three studies? Both timing of prime following arrival and timing of boost differed between Study A and B as well as the time of measurement of the immunoglobulin response following the second vaccination. For study A, why were antibodies measured 2,5 weeks after vaccine dose 1 but 3,5 weeks after vaccine dose 2

Line 145 and 151, describe when antibody titers were measured in relation to vaccination

Stud C, Line 153, antibody titers are measured 1 week following vaccination, why was this time point chosen? Why was a different interval chosen for duration of diet and antibiotics in relation to stud a and c?

Line 233 – How were titers accounted for in the statistics, why were antibody titers not transformed? Please give rationale or log-transform antibodies/present as geometric mean titers

Lines 258/262, list test used for significance

Line 259 – please define TM7

Line 263 – Phylum should not be italicized

Line 269, which microbiome metric is being given a significance here?

Line 272, what was the impact of antibiotics on weight in study C

Line 276, missing figure legend for figure 1?

Figure 2, e and f, list time of blood draw in relation to vaccination; list timing of fecal sample collection, list timing of vaccination in figure or figure legend. Correct figure label for e/f to be anti-tetanus IgG

Figure 3, list timing of fecal sample collection for microbiome, list timing of IgG in relation to vaccination, list timing of vaccination. Correct figure legends to be anti-tetanus IgG

Line 280/285/294 –is this absolute or relative abundance?

Figure 4e, significance is listed as text but not shown for antibiotics – what has a significance of p= 0.008

Table 2, describe duration of time mice were fed diet (4,5 weeks from Study C or 2 weeks from Study A?) Add n for column control and gluten-free

Line 309, figure 1 does not show a serum sample 2 weeks after the first vaccination for Study B, please adjust.

Line 312, what data/statistical test supports the borderline lower variation. Please list statistical test for all p-values.

Line 316, please indicate which comparison is being made for the significance here

Lines 321-337, please do not report trends, only significant values

Figure 5, please indicate which comparisons are significant in the figure rather than describing them on the figure. Please indicate how many mice were specifically in each group (e.g. n=10/group, not 10 overall?). Please label the individual plots, rather than 5a/b alone. Figure 5B seems to be missing?

Line 367, suggest to specify IgG with anti-tetanus IgG

Line 368, remove reference to ‘low-responder’ unless this term is defined and tested with a statistical bound.

Line 369, suggest to remove dramatic, suggest to specify ‘reduced anti-tetanus IgG following the first tetanus vaccination”

Line 372, inter-individual variation in GM or antibody response?

Lines 372-374, this sentence’s meaning is confusing

Line 383, these findings were associative, do the authors mean, for future mechanistic explanations?

Line 384/393, suggest correlating bacterial abundance to antibody titer/Treg to support such an association.

Line 395, see earlier comment on definition high/low responder

Please provide a limitations section, in which differences in experimental design between the study arms are taken into account. Also please address the limitation of doing this study in mice and looking only at tetanus vaccine. Please address the potential impact of the metabolome as mediator of a microbiome effect as well as other components of the microbiome, such as virome/fungiome that have not currently been measured.

7. PLOS authors have the option to publish the peer review history of their article (what does this mean?). If published, this will include your full peer review and any attached files.

Reviewer #2: No

Reviewer #3: No

---

## [Author Response · Author response to Decision Letter 1]

27 Jan 2022

Our point-to-point replies are as follows:

Reviewer #2: 

Fig5 is problematic, it is not known which two groups are different from each other simply by showing a p-value. The same thing is true for Fig4a,b, and e.

We understand the concerns of the reviewer, because it is very common to analyse such data by a one-way ANOVA followed by, what in this case would be a Tukey’s post hoc test. However, if the study as in this case is planned accordingly the most powerful way to analyse these data are by a two-way ANOVA, and in a two-way ANOVA you get a p-value for each factor across the groups. We do not find that it is statistically appropriate (although offered by some statistical software packages) to supply it with a post-hoc test, as the beauty of a two-way ANOVA is that you get one p-value for each factor, and attempts to calculate more p-values increases the risk of type I errors without actually adding any value to the evaluation. In addition, the p-values to these factors fully respond to the hypothesis. However, if the editor prefers that we do it, we will of course do it and put it into the table.

Ampicillin treatment in study B is a little bit too long. Does short-term one-week treatment also have an impact?

We are a bit uncertain with how the reviewer defines ‘too long’, but we understand it as if it should be compared to a human situation. In this study ampicillin treatment was used to block microbiota impact on the vaccine response to show if there was a microbiota related effect. It is common practice in antibiotic depletion studies to start out by a depletion for the entire study period, i.e. in this case during the vaccination and until the last antibody monitoring. So, this is what we have done. A short term depletion changes the microbiota afterwards, and we are in the process of studying whether such a change can have long term effects. We do not at present have the basis for further studies on such short term treated animals, and more basic studies of potential immune changes are needed to make out, when to expect changes in a vaccine response. However, we admit that our studies cannot be used to state anything about how the general use of ampicillin/amoxicillin for children affects their immune response; especially not if they are not treated with it at vaccination time point. This should have been better discussed, and we have now introduced the following in the discussion:

L430-438: In this study, mice were treated with ampicillin throughout the study, which is much longer than humans normally would be treated. During ampicillin treatment the majority of naturally gut inhabiting bacteria will either be eradicated or heavily suppressed, and when ampicillin treatment in mice is terminated, some microorganisms will recover from the host, while other will recover from the environment, and, therefore, ampicillin treated mice will have a microbiota different from non-treated mice [31]. It is unknown whether such a microbiota change will have long term impact on the immune system, and, therefore, our studies cannot be used to state whether the general use of antibiotics in children hampers their responses to vaccines; especially not if they are not treated during vaccination.

We have also explained this better at the end of the introduction:

LN99-100: To block the impact of the GM, some of the mice were treated or not treated with ampicillin.

Can the author demonstrate that the gluten-free diet-induced microbiota changes drive the lower primary antibody response by fecal transfer type of experiment? 

We agree that this I a perspective for further studies, which has already been mentioned in the discussion (‘To verify this further studies should attempt to transfer the GM effect to germ-free mice through a fecal matter transplantation’). We have a manuscript accepted for Journal of Autoimmunity, in which we in a two-generation show that a gluten-free diet effect is transferred over generations, and some of the effect is microbiota dependent and some is microbiota independent. We have inserted a reference to this:

L383-393: In humans, it has been shown that gluten-free diet induced GM changes are essential for the increased Treg levels [51], although also a direct anti-inflammatory effect of a gluten-free diet has previously been shown in mice [52]. From studies in two-generation studies in non-obese diabetic mice, it is known that the effects of a gluten-free diet on the immune system can be transferred over generations both dependently and indepently of the GM [15, 53]. In the short term study C it was difficult to verify any major anti-tetanus IgG titer effect of the gluten-free diet one week after vaccination, and the impact of the gluten-free feeding was also minor compared to the two other studies. However, there was still in terms of cell counts a clear impact on the immune system, which therefore either responds to even small GM changes and/or directly to the changed diet composition. Whether the impact on vaccine response is GM dependent might be further elucidated through a fecal matter transplantation to germ-free mice. 

Can the author demonstrate which cell population influenced by the diet reduced the antibody response?

FACS data were generated on study C mice, which was too short to show the immediate effects of the first vaccination, as this was when a difference was observed. However, this should have been mentioned in the discussion. We have now written:

LN387-389: In the short term study C it was difficult to verify any major anti-tetanus IgG titer effect of the gluten-free diet one week after vaccination, and the impact of the gluten-free feeding was also minor compared to the two other studies.

The language can be modified further. p10 L223, please delete the ' mark.

We have made sure that there is no such mark in L223. We have gone through the manuscript with the department translator, and it has be substantially changes. If this from an English language point of view is not enough, we will after acceptance pass it through an editorial service if requested by the editor.

Reviewer #3: 

The article aims to characterize the impact of microbiome alteration (either through diet or antibiotics) on tetanus vaccine immunogenicity in mice. The article is a nice addition to the literature that suggests that alteration of the intestinal microbiome alters host immune response to both vaccines and pathogens. The authors have performed experiments with a large sample sizes and evaluated microbiome, immune cells, and gene expression resulting in an interesting data set that suggests gluten free-diet and antibiotic administration diminish anti-tetanus antibodies following intial vaccination, possibly through different mechanisms.

Major comments

The reviewer requests a better rationale for why tetanus vaccine was used for this study, given that it is known to elicit consistently high antibody responses across geography and across ages.

This is a quite obvious and understandable request from the reviewer. When we planned the study, , we found – as we tried to relate the effect to bacteria of the gut – that it would be most relevant in the first place to study this in a vaccine based upon a bacterial product known to induce a strong Th1 reaction, and therefore we chose the tetanus vaccine. Obviously, in the light of COVID-19 pandemic, it would probably have been relevant to base our studies on a virus vaccine. Having said that we can also see from some of the work we do today that the responses we see in relation to viral vaccines is stronger in their CD8 than in their CD4 reaction, and the knowledge we have built from our work with gluten-free diets over the last decade points in the direction of a gluten-free impact on CD4 rather than CD8. We have changed the last part of the introduction into:

LN90-100: We hypothesized that gluten-free diet induced GM perturbations would influence vaccine response in mice. We found it most relevant to study this in relation to a vaccine based upon a bacterial product, and, therefore, we chose to study the impact on a tetanus toxoid vaccine, although this vaccine generally have high immunogenicity and protection globally [39]. Tetanus toxoid is a bacterial product, which when used as vaccine, is known to raise a strong Th reaction [40], and it has traditionally been used with aluminium oxide hydrate adjuvant, which is expected to drive a Th2 rather than a Th1 response [41]. An increase in Treg relative abundance and subsequent IL-10 secretion, as induced by a gluten-free diet, will reduce both Th1 and Th2 and, thereby antibody producing B cell relative abundance [24, 42]. Therefore, we used the production of specific IgG as primary readout in mice fed a gluten-free or a standard gluten containing diet. To block the impact of the GM, some of the mice were treated or not treated with ampicillin.

We have also touched upon it in the limitations:

LN463-471: Finally, these studies were performed with a tetanus toxoid vaccine, which globally in humans is known as a highly efficient vaccine [39], and to which the immune system responds in ways, which can be different from responses to other vaccines. The effectiveness of viral vaccines seems to vary far more. Titers to influenza varies significantly between vaccinated humans in different geographical regions [1], and even the most efficient COVID-19 vaccines do not offer full protection against infection in humans [74]. Therefore, our studies in mice should not be used to question the efficiency of tetanus vaccination in humans, and from a more applied point of view it would be equally or more relevant to study the impact of the GM on such vaccines; especially in the light of the COVID-19 pandemic.

Also suggest to reference more of the recent (human and murine) literature on how microbiota alters vaccine immunogenicity.

Having gone through the references we admit that some of these were somewhat old, and that certain subjects could be better elucidated by some more recent references, which we have now included in the manuscript.

Study design - Studies A, B and have differing intervals of treatment (antibiotic/diet) and differing timing of prime and boost of vaccination as well as measurement of antibody responses making it difficult to extrapolate results across the studies. The rationale for choices are not clearly described.

There are some differences, but the way we have presented the study setups make it look more different than it is. In study A and study B mice were vaccinated at 8 weeks of age, but in the previous manuscript we wrote, when they were vaccinated in relation to the arrival. This is, of course, relevant in relation to how long they were fed gluten-free diet or antibiotics, but this was mostly decided by the vendors delivery plans and not by biological considerations. Normally when we do long term feeding/drinking water intake experiments we put the mice on the diet/water at arrival to avoid stressful changes more than once. After study A we discussed whether 2,5 weeks between the two vaccinations was too short and we decided to use four weeks instead. In study C, when we did the vaccinations only once, we wanted to be sure to have mice with a full functioning immune system and, therefore, we decided to give them an extra week before vaccination.

We have introduced the following under materials and methods:

LN 129-137: We set up three studies in BALB/cBomTac mice, to investigate the influence of gut microbiota manipulations on the response to a tetanus vaccine. Two studies (A and B) to study the impact of either a gluten-free diet or antibiotics on the antibody production after a boosted vaccination regime, and one study (C) to study the immediate impact of one vaccination on immune cells counts in gluten-free fed or antibiotics treated mice. In study A we boosted with a second vaccination after two and a half weeks, while we in study B reconsidered this and did the boosting after four weeks. In study C we used only one vaccination, because the aim was to see the cell counts in relation to our observations after the first vaccination in study A and B, and, therefore, we also sampled the animals only one week later.

Analysis – Given the use of antibody titers, request that the authors repeat antibody statistical analyses using transformed data/geometric mean titers.

We have ranked all titer values and re-analyzed them. Outcomes were not dramatically changed. Graphs have been changed to show medians instead of means.

Article can often be confusing to read and suggest input from a native English speaker. Some hypotheses in introduction and conclusion suffer from large jumps across subjects without clear delineation of whether data comes from human or animal subjects. Suggest to review and adjust claims accordingly.

Reading the manuscript again we fully agree. We have been thoroughly through the paper together with the department translator, and the introduction and the discussion have both been essentially reorganized. If this from an English language point of view is not enough, we will after acceptance pass it through an editorial service if requested by the editor.

Minor comments

Abstract

Line 26 Wording is confusing, isn’t the primary purpose of the study to evaluate the effect of diet ( gluten-free or gluten-containing) and antibiotics (ampicillin or not) on tetanus vaccine immunogenicity? Suggest to adjust wording

Point taken. It now reads:

LN26-27: The purpose of this study was to compare the effect of a tetanus vaccine on immunoglobulin G titers and immune cell levels in female BALB/c mice fed either a gluten-free or a gluten containing diet and treated or not treated with ampicillin.

Line 28 suggest change ‘lowered’ to ‘was associated with’

Done

Line 29 low-responder not defined in abstract or text, suggest to define if this is a statistical outcome or otherwise remove

All references to ‘Low responders’ have been removed.

Line 29 relative abundance or abundance?

It no reads ‘the relative abundance’

Line 30 remove word massively, change to significantly if this is the case. If hypothesis is that diet altered microbiota which then altered vaccine response, suggest to start with change in microbiota in this sentence

‘Massively’ has been replaced with ‘significantly’

Line 30, suggest to remove ‘similarly’- suggests that antibiotics altered microbiota similarly to diet?

‘Similarly’ has been replaced with ‘also’.

Line 31, add significance for all findings

‘Significantly’ has been inserted

Line 31, define ‘boosting’

‘Boosting’ has been replaced with ‘a second vaccination’

Line 32, what was definition of” influenced’’?

‘Influenced’ has been replaced with ‘reduced’

Line 36, suggest to alter sentence, “these results suggest …..”

Done

Introduction

Line 44 The authors describe diminished vaccine efficacy in low-middle income settings, however tetanus vaccines have never been shown to have diminished efficacy in low or middle income settings. Rather, they have high immunogenicity and good protection globally, with possible higher immunogenicity in low-income settings. Please correct

This is a general statement, which is indeed true for some vaccines. We have later in the introduction explained our rationale for choosing the tetanus vaccine, and we have inserted a sentence describing that it is generally efficient:

LN91-98: We found it most relevant to study this in relation to a vaccine based upon a bacterial product, and, therefore, we chose to study the impact on a tetanus toxoid vaccine, although this vaccine generally have high immunogenicity and protection globally [39]. Tetanus toxoid is a bacterial product, which when used as vaccine, is known to raise a strong Th reaction [40], and it has traditionally been used with aluminium oxide hydrate adjuvant, which is expected to drive a Th2 rather than a Th1 response [41]. An increase in Treg relative abundance and subsequent IL-10 secretion, as induced by a gluten-free diet, will reduce both Th1 and Th2 and, thereby antibody producing B cell relative abundance [24, 42].

Line 46 Suggest to use the term low-middle income and high income and not ‘developing”and developed”as terminology

Good point. Done.

Line 48 – all citations have to do with changing vaccine immunogenicity by genetics and age and do not support differential vaccine performance by geography. Please correct

The references have been replaced

Line 71, remove the word óbviously’, suggest to make less definitive statements, add “can cause” and add that this study was in mice

The sentence now reads:

LN79: Antibiotics, such as ampicillin, may have a strong impact on GM, which causes decreased abundances of Treg and Th in both mice [31] and humans [32].

Line 74, modify that this is a study that was performed in mice

Done

Line 76, add that this is for influenza vaccination

Done

Line 83, cite literature that shows this

Done

Line 88, citation is not about Bangladeshi children

This has been corrected

Study Design

Line 136, what was the rationale for the different timing of tetanus vaccination between the three studies? Both timing of prime following arrival and timing of boost differed between Study A and B as well as the time of measurement of the immunoglobulin response following the second vaccination. For study A, why were antibodies measured 2,5 weeks after vaccine dose 1 but 3,5 weeks after vaccine dose 2

This has been clarified and corrected above.

Line 145 and 151, describe when antibody titers were measured in relation to vaccination

Stud C, Line 153, antibody titers are measured 1 week following vaccination, why was this time point chosen? Why was a different interval chosen for duration of diet and antibiotics in relation to stud a and c?

This has been clarified in a previous comment to the reviewer. We have revised the study plan descriptions accordingly.

Line 233 – How were titers accounted for in the statistics, why were antibody titers not transformed? Please give rationale or log-transform antibodies/present as geometric mean titers

Antibody titers were calculated according to the standard curve made up according to the suppliers instructions. This has been put into materials and methods (LN204). The statistics have been recalculated as described above.

Lines 258/262, list test used for significance

The tests have already been described in the statistics section.

Line 259 – please define TM7

The bacterium used to have the candidate name TM7, but it has been reclassified as Saccharibacteria_ Saccharimonadaceae. This has been put as a footnote in the table.

Line 263 – Phylum should not be italicized

Corrected

Line 269, which microbiome metric is being given a significance here?

Relative abundances. This has been written in the legend now.

Line 272, what was the impact of antibiotics on weight in study C

The mice were not weighed at the end of this study.

Line 276, missing figure legend for figure 1?

It has now been inserted (LN127).

Figure 2, e and f, list time of blood draw in relation to vaccination; list timing of fecal sample collection, list timing of vaccination in figure or figure legend. Correct figure label for e/f to be anti-tetanus IgG

Done

Figure 3, list timing of fecal sample collection for microbiome, list timing of IgG in relation to vaccination, list timing of vaccination. Correct figure legends to be anti-tetanus IgG

Line 280/285/294 –is this absolute or relative abundance?

All ‘abundance’ have been replaced with ‘relative abundance’.

Figure 4e, significance is listed as text but not shown for antibiotics – what has a significance of p= 0.008

This is a P-value calculated in a two-way ANOVA. The test gives a P-value for each factor, i.e. antibiotics+/- and gluten-free/gluten. The P-value for the factor antibiotics+/- is 0.008 (actually after ranking the data it is now 0.002). There is no significant effect of a gluten-free diet. We cannot really see other ways to write this, and this is also how we normally write the outcome of a two-way ANOVA in papers.

Table 2, describe duration of time mice were fed diet (4,5 weeks from Study C or 2 weeks from Study A?) Add n for column control and gluten-free

Done

Line 309, figure 1 does not show a serum sample 2 weeks after the first vaccination for Study B, please adjust.

This has been corrected.

Line 312, what data/statistical test supports the borderline lower variation. Please list statistical test for all p-values.

This was shown by Levene’s test, which has now been stated.

Line 316, please indicate which comparison is being made for the significance here

A two-way ANOVA on ranked data, which has now been stated.

Lines 321-337, please do not report trends, only significant values

Point taken. It has been omitted.

Figure 5, please indicate which comparisons are significant in the figure rather than describing them on the figure. 

As also stated for reviewer No 2, these P-values have been calculated in a two-way ANOVA, in which you get a P-value for each factor. We do not consider it appropriate statistics to do post-hoc tests after a two-way ANOVA, as the increased search for P-values increases the risk of type I error, and the P-values generated by the two-way ANOVA fully responds to the hypothesis. However, if the editor wants us to do it, we will do it and put it into the figures.

Please indicate how many mice were specifically in each group (e.g. n=10/group, not 10 overall?). 

Done

Please label the individual plots, rather than 5a/b alone. 

The individual plots are labelled.

Figure 5B seems to be missing?

We are not aware in which part of the process this problem has been generated, but it has been submitted with this revision.

Line 367, suggest to specify IgG with anti-tetanus IgG

Done

Line 368, remove reference to ‘low-responder’ unless this term is defined and tested with a statistical bound.

Done

Line 369, suggest to remove dramatic, suggest to specify ‘reduced anti-tetanus IgG following the first tetanus vaccination”

Done

Line 372, inter-individual variation in GM or antibody response?

This phrase has been deleted from the restructured discussion.

Lines 372-374, this sentence’s meaning is confusing

We agree, and it has disappeared in the revised discussion.

Line 383, these findings were associative, do the authors mean, for future mechanistic explanations?

Yes, we have now written for future mechanistic studies.

Line 384/393, suggest correlating bacterial abundance to antibody titer/Treg to support such an association.

We have correlated all OTU’s of the gut microbiota sequencing with the IgG, and it gave a little that we have inserted as a S3 Table. We are not convinced that it makes sense to highlight it very much in the paper.

We write as follows:

LN273-277: We found a significant correlation after FDR correction of especially species within the phylum Bacteroidetes with the pooled IgG levels of both time points in the antibiotic treated mice compared to their controls (Study B), while no correlations were found after the first vaccination, in singe groups or in study A (S3 Table). 

LN406-409: We observed a correlation with anti-tetanus IgG in study B, i.e. the antibiotics treated mice. However, this should not be over-interpreted, because in the antibiotics treated group, which had a reduced anti-tetanus IgG production, there was reduced relative abundances of Bacteroidetes, and we found no correlations in study A.

Line 395, see earlier comment on definition high/low responder

We have omitted all talk about high and low responders.

Please provide a limitations section, in which differences in experimental design between the study arms are taken into account. Also please address the limitation of doing this study in mice and looking only at tetanus vaccine. 

We have inserted a limitations section after the discussion, and removed the last part of the discussion to a conclusions section:

LN458-471: Studies in mice typically give some basic information, but they are quite often not translatable to humans, i.e. our data cannot be taken as a solid statement that a gluten-free diet has the same impact in humans. Also, there were differences in the experimental setup when testing the impact of a gluten-free diet and the impact of antibiotics. So, although it seemed as if the way these two factors impacted the vaccine responses differed, this interpretation should be made with the precaution that the two studies are not directly comparable. Finally, these studies were performed with a tetanus toxoid vaccine, which globally in humans is known as a highly efficient vaccine [39], and to which the immune system responds in ways, which can be different from responses to other vaccines. The effectiveness of viral vaccines seems to vary far more. Titers to influenza varies significantly between vaccinated humans in different geographical regions [1], and even the most efficient COVID-19 vaccines do not offer full protection against infection in humans [74]. Therefore, our studies in mice should not be used to question the efficiency of tetanus vaccination in humans, and from a more applied point of view it would be equally or more relevant to study the impact of the GM on such vaccines; especially in the light of the COVID-19 pandemic.

Please address the potential impact of the metabolome as mediator of a microbiome effect as well as other components of the microbiome, such as virome/fungiome that have not currently been measured.

We have now inserted the following in the discussion:

LN439-445: The dietary impact on the microbiome not only relates to the bacterial community of the host. Even specific pathogen free mice harbours a gut virome both in the form of phages [64] and mammal viruses [65]. These may have a major impact on host responses [66]. The microbiomic as well as the host related digestion produces a metabolome, and the components of this may equally well facilitate differences in immune responses [67].

---

## [Decision Letter · Decision Letter 2]

28 Feb 2022

PONE-D-21-12086R2Dietary Gut Microbiota Perturbations Influence Murine Vaccine ResponsePLOS ONE

Dear Dr. Kornerup Hansen,

Thank you for submitting your manuscript to PLOS ONE. After careful consideration, we feel that it has merit but does not fully meet PLOS ONE’s publication criteria as it currently stands. Therefore, we invite you to submit a revised version of the manuscript that addresses the points raised during the review process.

 While some of the previous issues raised by the reviewers were addressed, there remain a number of concerns that still need to be more adequately addressed. In particular, the concerns of reviewer 2 regarding the immune cell responses (figure 5) and gluten-free diets are still valid and should be better addressed.

We look forward to receiving your revised manuscript.

Kind regards,

Brenda A Wilson, Ph.D.

Academic Editor

PLOS ONE

Reviewers' comments:

Reviewer's Responses to Questions

**Comments to the Author**

1. If the authors have adequately addressed your comments raised in a previous round of review and you feel that this manuscript is now acceptable for publication, you may indicate that here to bypass the “Comments to the Author” section, enter your conflict of interest statement in the “Confidential to Editor” section, and submit your "Accept" recommendation.

Reviewer #2: (No Response)

Reviewer #3: (No Response)

2. Is the manuscript technically sound, and do the data support the conclusions?

Reviewer #2: Partly

Reviewer #3: Yes

3. Has the statistical analysis been performed appropriately and rigorously? 

Reviewer #2: No

Reviewer #3: Yes

4. Have the authors made all data underlying the findings in their manuscript fully available?

Reviewer #2: Yes

Reviewer #3: Yes

5. Is the manuscript presented in an intelligible fashion and written in standard English?

Reviewer #2: Yes

Reviewer #3: Yes

6. Review Comments to the Author

Reviewer #2: I am not satisfied with the authors response.

The examination of the immune cell responses in Figure 5 needs to be clarified a little bit more.

1) what is the baseline levels of all the immune cell population without vaccination?

2) why the n number is not consistent (e.g., for Treg n=5, while total CD3 T n=10?)

3) how to explain the cell number change differences between spleen and MLN?

4) Pls indicated which two groups were compared when indicated there is a statistcial significance.

In my opinion, a fecal transfer experiment to demonstrate the effect of gluten-free diet on IgG response is dependent on gut microbiota should be performed to strength this manuscript.

Reviewer #3: The authors have addressed the majority of my comments and questions

Small comments:

with reference to the revised track changes text

(page 54) Abstract: line 32, add lower initial vaccine titer

(page 58) line 114: although this vaccine generally has high immunogenicity….

(line 161) ‘we reconsidered this’, suggest to include a rationale here

(lines 176/ 183) body weights of mice … were monitored. What was the impact of antibiotics and diet on weight in studies A/B (this could be confounding)

(line 314/315) describe the direction of the correlation and clarify if correlation was for the phylum or species level.

Line 317 typo, ‘single’

Line 347, Table 2 – list all FDRs in table or indicate which FDR level was used to generate table. Relative repeated twice in table legend

Please review text for typos

7. PLOS authors have the option to publish the peer review history of their article (what does this mean?). If published, this will include your full peer review and any attached files.

Reviewer #2: **Yes: **Yugang Wang

Reviewer #3: No

---

## [Author Response · Author response to Decision Letter 2]

3 Mar 2022

Reviewer #2: 

The examination of the immune cell responses in Figure 5 needs to be clarified a little bit more.

1) what is the baseline levels of all the immune cell population without vaccination?

4) Pls indicated which two groups were compared when indicated there is a statistcial significance.

With due respect to the reviewer, we think that the reviewer has misunderstood the experimental setup. The hypotheses decided upon during the planning of the study were (1) ‘Does gluten have an impact on immune cell populations?’ and (2) ‘Does antibiotics have an impact on immune cell populations’? It is standard within experimental design to answer such two-factorial research questions in a two-factorial setup evaluated by a two-way ANOVA. The set-up looks as follows:

 Factor 1 Gluten

 + -

Factor 2 + Group 1 Group 3

Antibiotics - Group 2 Group 4

The two-way ANOVA will give an individual p-value for each of the factors. Of course, the groups can be compared in a post-hoc test, such as Tukey’s post hoc comparison, but we have no hypothesis relating to differences between the individual groups and the study was not planned for this. The only statistical effect will be that we get more p-values, which we subsequently will have to create hypotheses to explain. This is not good experimental design and statistics.

Alternatively, the study could have been set up for evaluation by a one-way ANOVA, i.e. an unvaccinated group to be compared with different groups with or without gluten and antibiotics. If an overall significant p-value was found in the one-way ANOVA these groups could subsequently be compared with the control group in a Dunnett’s post hoc test. This is, in our eyes, a much weaker design, because it is difficult to interpret the effects of the two factors individually, and it implies multiple testing, which is FDR corrected in the Dunnett’s post hoc test. 

However, no matter, which design is preferred, the simple answer is that we planned and did the two-way setup and this cannot subsequently be changed into a one-way setup.

As mentioned in our previous rebuttal letter, if the editor wants us to do the post-hoc test, it can, of course, be done, but in our eyes, it still would not be correct. However, the data in Figure 5 is a minor part of the study, and we do not want them to be a blocker for publication in PLOS One, so we will do as he editor prefers.

2) why the n number is not consistent (e.g., for Treg n=5, while total CD3 T n=10?)

For rationalization, we chose to limit the number of mice tested for some of the antibodies. A few cell preparations were unsuccessful, and for those few groups n was reduced accordingly. It was already explained in the previous version of the manuscript in line 226-228: ‘For rationalization some antibodies were only tested on randomly selected mice, and 5, 10 or 15 mice were tested on each level (Figure 5a-b). In some groups some of the preparations were unsuccessful and, therefore, n was reduced accordingly.’

3) how to explain the cell number change differences between spleen and MLN?

The mesenteric lymph nodes represent the immune reaction locally in the gut, while the spleen is regarded as an expression of the systemic response. Not all reactions in the gut are translated into a systemic response. This was already explained in line 396 – 399 in the previous version of the manuscript: ‘In the present study we found that the gluten-free diet lowered the levels of Treg locally in the gut, while in the spleen, which should be regarded as an expression of the systemic effect, the gluten-free diet increased Treg fractions, CD4+ T cell activation, and tolerogenic dendritic cell fractions and activation, thereby extending the downregulating effect of the Treg’.

In my opinion, a fecal transfer experiment to demonstrate the effect of gluten-free diet on IgG response is dependent on gut microbiota should be performed to strength this manuscript.

The reviewer has a point but there is already a large data amount in the present paper, and the paper has been on a long track with the journal. We have already mentioned this in the discussion in the previous version of the manuscript (ln 390-392): ‘Whether the impact on vaccine response is GM dependent might be further elucidated through a fecal matter transplantation to germ-free mice.’ We also refer to another study we have published, while awaiting the progression of the present paper, showing how immune effects of a gluten-free diet can be both microbiota dependent and independent. We are in a process of doing further studies within this subject, and fecal matter transplantation is, of course, relevant. However, we also acknowledge the input from reviewer 3 that we should consider other infectious agents, and, therefore, it is not relevant to put these data into this manuscript.

Reviewer #3

The authors have addressed the majority of my comments and questions

Thank you.

Small comments:

(page 54) Abstract: line 32, add lower initial vaccine titer

Done

(page 58) line 114: although this vaccine generally has high immunogenicity….

Done

(line 161) ‘we reconsidered this’, suggest to include a rationale here

The sentence now reads’ while we in study B considered that his might be too short and did the boosting after four weeks’.

(lines 176/ 183) body weights of mice … were monitored. What was the impact of antibiotics and diet on weight in studies A/B (this could be confounding)

This is the ‘materials and methods’ section, and therefore probably it is not the right place to give results. However, the weight curves are shown in S1 Figure and the interpretation that both diet and antibiotics influence weight are referred to in the first sub-section of ‘Results’ (ln 270-273): ‘Mice fed the Altromin modified gluten-free diet weighed significantly more than mice fed the standard Altromin diet in study A (S1 Figure a, P = 0.000). Antibiotics in the drinking water reduced the weight significantly in study B (S1 Figure b, P = 0.001), but this was mostly due to a weight decrease in the first weeks on antibiotics.’

(line 314/315) describe the direction of the correlation and clarify if correlation was for the phylum or species level

.

Done

Line 317 typo, ‘single’

Done

Line 347, Table 2 – list all FDRs in table or indicate which FDR level was used to generate table. Relative repeated twice in table legend

Done

---

## [Decision Letter · Decision Letter 3]

9 Mar 2022

PONE-D-21-12086R3Dietary Gut Microbiota Perturbations Influence Murine Vaccine ResponsePLOS ONE

Dear Dr. Kornerup Hansen,

Thank you for submitting your manuscript to PLOS ONE. After careful consideration, we feel that it has merit but does not fully meet PLOS ONE’s publication criteria as it currently stands. Therefore, we invite you to submit a revised version of the manuscript that addresses the points raised during the review process.

 Both reviewers agreed that the revised manuscript was significantly improved. However, Reviewer 2 still has some concerns about the study design and analysis, which really could only be addressed by substantial additional experimentation, which I agree is probably outside the scope of the study since Figure 5 is not a large portion of the overall study. In lieu of that, a title that reflects the manuscript content better and inclusion of additional statistical analysis, as per the reviewer's suggestion, could help address this. But, it will be important to include additional comparative discussion of the results, including caveats and limitations of the study design in light of the results and perhaps suggestions for future studies.

We look forward to receiving your revised manuscript.

Kind regards,

Brenda A Wilson, Ph.D.

Academic Editor

PLOS ONE

Journal Requirements:

Additional Editor Comments (if provided):

Reviewer 2 still has some issues with the authors' responses regarding Figure 5 and the goals of the study. I think some of this could be addressed by changing the title to be more reflective of the manuscript content (as is, it is a bit sweeping in nature). I suggest something on the order of: "Effect of gluten-free diet and antibiotics on murine gut microbiota and immune response to tetanus vaccination." I think it would also be good for the authors' to include the Dunnett's post hoc test (in addition to what has already been done) and provide a brief comparative discussion of the results, including caveats to the analyses and limitations of the study design.

Reviewers' comments:

Reviewer's Responses to Questions

**Comments to the Author**

1. If the authors have adequately addressed your comments raised in a previous round of review and you feel that this manuscript is now acceptable for publication, you may indicate that here to bypass the “Comments to the Author” section, enter your conflict of interest statement in the “Confidential to Editor” section, and submit your "Accept" recommendation.

Reviewer #2: (No Response)

Reviewer #3: All comments have been addressed

2. Is the manuscript technically sound, and do the data support the conclusions?

Reviewer #2: No

Reviewer #3: (No Response)

3. Has the statistical analysis been performed appropriately and rigorously? 

Reviewer #2: Yes

Reviewer #3: (No Response)

4. Have the authors made all data underlying the findings in their manuscript fully available?

Reviewer #2: Yes

Reviewer #3: (No Response)

5. Is the manuscript presented in an intelligible fashion and written in standard English?

Reviewer #2: Yes

Reviewer #3: (No Response)

6. Review Comments to the Author

Reviewer #2: 1) The authors claim that "Gut Microbiota Perturbations Influence Murine Vaccine Response", but have no intentions to prove their conclusions by providing more experimental data.

2) The authors claim that CD3 T cells are reduced by feeding with a Gluten-free diet in Fig 5A, however, it is really hard to believe that it is a biologically significant reduction at least for the "vacc GF" group, although statistically maybe there is a difference. The two-way ANOVA analysis is mathematically correct, but the way the authors presented it in the current format is hard to interpret.

Reviewer #3: (No Response)

7. PLOS authors have the option to publish the peer review history of their article (what does this mean?). If published, this will include your full peer review and any attached files.

Reviewer #2: No

Reviewer #3: No

---

## [Author Response · Author response to Decision Letter 3]

11 Mar 2022

We have updated the reference:

53. Hansen CHF, Larsen CS, Zachariassen LF, Mentzel CMJ, Laigaard A, Krych L, et al. Gluten-free diet reduces autoimmune diabetes mellitus in mice across multiple generations in a microbiota-independent manner. J Autoimmun. 2022;127:102795. Epub 2022/02/02. doi: 10.1016/j.jaut.2022.102795. PubMed PMID: 35101708.

Reviewer 2 still has some issues with the authors' responses regarding Figure 5 and the goals of the study. I think some of this could be addressed by changing the title to be more reflective of the manuscript content (as is, it is a bit sweeping in nature). I suggest something on the order of: "Effect of gluten-free diet and antibiotics on murine gut microbiota and immune response to tetanus vaccination." 

We have changed the title to: ‘Effect of gluten-free diet and antibiotics on murine gut microbiota and immune response to tetanus vaccination’.

I think it would also be good for the authors' to include the Dunnett's post hoc test (in addition to what has already been done) and provide a brief comparative discussion of the results, including caveats to the analyses and limitations of the study design.

We have done Dunnett’s post hoc comparisons on all data in Figure 5 comparing the factorial groups to the ‘Vacc’ group. We have explained this in the statistics section and in the figure legend. We have introduced a sentence under ‘Limitations’: ‘The two-way setup for testing the impact on cell counts does not include a non-vaccinated group, so it is only possible to make conclusions on the relative impact of the experimental factors.’

---

## [Editor Report · Decision Letter 4]

28 Mar 2022

Effect of gluten-free diet and antibiotics on murine gut microbiota and immune response to tetanus vaccination

PONE-D-21-12086R4

Dear Dr. Kornerup Hansen,

We’re pleased to inform you that your manuscript has been judged scientifically suitable for publication and will be formally accepted for publication once it meets all outstanding technical requirements.

Kind regards,

Brenda A Wilson, Ph.D.

Academic Editor

PLOS ONE
---

## [Editor Report · Acceptance letter]

4 Apr 2022

PONE-D-21-12086R4 

Effect of gluten-free diet and antibiotics on murine gut microbiota and immune response to tetanus vaccination 

Dear Dr. Kornerup Hansen:

I'm pleased to inform you that your manuscript has been deemed suitable for publication in PLOS ONE. Congratulations! Your manuscript is now with our production department. 

Kind regards, 

on behalf of

Dr. Brenda A Wilson 

Academic Editor

PLOS ONE